# Stopping Bayesian Optimization with Probabilistic Regret Bounds

**James T. Wilson**
Morgan Stanley, New York, USA
`james.t.wilson@morganstanley.com`

## Abstract

Bayesian optimization is a popular framework for efficiently tackling black-box search problems. As a rule, these algorithms operate by iteratively choosing what to evaluate next until some predefined budget has been exhausted. We investigate replacing this de facto stopping rule with criteria based on the probability that a point satisfies a given set of conditions. We focus on the prototypical example of an $(\epsilon, \delta)$-criterion: *stop when a solution has been found whose value is within $\epsilon > 0$ of the optimum with probability at least $1 - \delta$ under the model.* For Gaussian process priors, we show that Bayesian optimization satisfies this criterion under mild technical assumptions. Further, we give a practical algorithm for evaluating Monte Carlo stopping rules in a manner that is both sample efficient and robust to estimation error. These findings are accompanied by empirical results which demonstrate the strengths and weaknesses of the proposed approach.

## 1   Introduction

In the real world, we are often interested in finding high-quality solutions to black-box problems. Many of these problems are not only expensive to solve but difficult to reason about without extensive background knowledge—such as discovering new chemicals [17], designing better experiments [48], or configuring machine learning algorithms [42].

A common approach is therefore to construct models for these problems and use them to predict real-world outcomes. In recent years, Bayesian optimization (BO) has emerged as a leading approach for accomplishing these tasks. Precise definitions vary, but BO methods are frequently characterized by their use of probabilistic models to guide the search for good solutions. The idea is for these models to provide distributions over the performance of competing alternatives, which can then be used to simulate the usefulness of evaluating different things. For a recent review, see Garnett [16].

Despite the success of these algorithms, an ongoing issue for practitioners has been the continued lack of interpretable stopping rules. The vast majority of BO runs proceed until a predetermined budget (e.g., a number of evaluations or amount of resources) is exhausted. We highlight two likely reasons for this trend and then give a brief prospective for model-based alternatives. The first reason is that stopping rules often revolve around quantities like optimums that are difficult to work with, even when defined under a model. The second is that even the best models sometimes go astray; and, if the model is bad, then model-based stopping is liable to stop much too soon or far too late. To avoid potential disappointment, let us say upfront that this work addresses the former challenge and only provides mild commentary on the latter. We will revisit this topic in the closing sections.

At the same time, we argue there is much to be gained by using models to help us decide whether a given solution is "good-enough" for its intended purpose [40]. One benefit of model-based stopping is its ability to adapt to the data. Sometimes, we will get lucky and stumble upon good solutions early on. Other times, our progress will be slow. If the model captures these events, then stopping can

38th Conference on Neural Information Processing Systems (NeurIPS 2024).

be tailored to each run. Another benefit of model-based stopping is its ability to simplify the user experience by asking us to specify what we wish to find instead of how much we wish to spend.

The basic idea we pursue is that, if we can simulate whether a solution is good-enough, then we can stop once we find one that probably is. We focus on a prominent example of this framework, but stress that much of what follows holds for different choices of models and conditions. In particular, we investigate the setting where the user deems a solution sufficient if its performance is within $\epsilon > 0$ of the optimum with probability at least $1 - \delta$ under the model.

Our primary contributions are to: i) combine recent work on scalable sampling techniques with algorithms for cost-efficient statistical testing; ii) show how the resulting estimators can be used as the basis for robust stopping rules; and, iii) introduce the first model-based stopping rule for BO with convergence and performance guarantees (up to model error).

The remaining text is organized as follows. Section 2 presents notation background material. Section 3 introduces the proposed stopping rule and evaluation strategy. Section 4 analyzes this algorithm's convergence and correctness. Finally, Section 5 investigates its empirical performance under idealized and realistic circumstances.

## 2  Background

We use boldface symbols to indicate vectors (lowercase) and matrices (uppercase). Given a sequence $(\boldsymbol{a}_i)$, we denote $\boldsymbol{a}_n = [a_1, \ldots, a_n]^\top$. Likewise, for a function $f : \mathcal{X} \to \mathbb{R}$, we use the shorthand $f(\mathbf{X}_n) = [f(\boldsymbol{x}_1), \ldots f(\boldsymbol{x}_n)]^\top$. By minor abuse of notation, we sometimes treat, e.g., $\mathbf{X}_n$ as a set.

We focus on the task of sequentially querying a function $f : \mathcal{X} \to \mathbb{R}$ in order to find a point $\boldsymbol{x} \in \mathcal{X}$ whose value $f(\boldsymbol{x})$ is within $\epsilon > 0$ of the supremum. Such a point is said to be $\epsilon$-*optimal* if this condition holds and $(\epsilon, \delta)$-*optimal* if it holds with probability at least $1 - \delta$. Throughout, we write $(\boldsymbol{x}_t)$ for the sequence of query locations.

At any given time $t \in \mathbb{N}_0$, our understanding of the target function's behavior is driven by domain knowledge and any data that we have already collected. We combine this information with the help of a Bayesian model by placing a prior on $f$ and defining an observation model. Different types of models are eligible and techniques introduced in the sequel simply require that we are able to simulate the chosen stopping conditions (e.g., $\epsilon$-optimality). We focus on the most popular family of models in this setting: Gaussian processes.

A Gaussian process (GP) is a random function $f : \mathcal{X} \to \mathbb{R}$ such that, for any finite set $\mathbf{X} \subseteq \mathcal{X}$, the random variable $f(\mathbf{X}) \in \mathbb{R}^{|\mathbf{X}|}$ is Gaussian in distribution. We write $f \sim \mathcal{GP}(0, k)$ for a centered GP with covariance $k : \mathcal{X} \times \mathcal{X} \to \mathbb{R}$ and model observations as function values corrupted by independent Gaussian noise, i.e. $y(\mathbf{X}_t) \mid f(\mathbf{X}_t) \sim \mathcal{N}\big(f(\mathbf{X}_t), \gamma^2 \mathbf{I}\big)$. Conditional on $y(\mathbf{X}_t)$, we therefore believe that $f$ is distributed as $f_t \sim \mathcal{GP}(\mu_t, k_t)$, where $\mathbf{\Lambda} = k(\mathbf{X}_t, \mathbf{X}_t) + \gamma^2 \mathbf{I}$ is used to define

$$\mu_t(\boldsymbol{x}) = k(\boldsymbol{x}, \mathbf{X}_t)\mathbf{\Lambda}^{-1}y(\mathbf{X}_t) \qquad k_t(\boldsymbol{x}, \boldsymbol{x}') = k(\boldsymbol{x}, \boldsymbol{x}') - k(\boldsymbol{x}, \mathbf{X}_t)\mathbf{\Lambda}^{-1}k(\mathbf{X}_t, \boldsymbol{x}'). \quad (1)$$

Finally, we assume that $\mathcal{X}$ is compact and that $\mu_t$ and $k_t$ are both continuous so their limits are attained on $\mathcal{X}$. Among other things, this assumption allows us to write $\boldsymbol{s}_t \in \arg\max_{\boldsymbol{x} \in \mathbf{S}_t} \mu_t(\boldsymbol{x})$ for a preferred solution at time $t$, where $\mathbf{S}_t$ is either the set of evaluated points $\mathbf{X}_t$ or the search space $\mathcal{X}$.

## 3  Method

Suppose Bayesian optimization terminates at time $t \in \mathbb{N}_0$ and returns a point $\boldsymbol{x} \in \mathcal{X}$ as the solution. Our *regret* for having returned this point is defined as the distance between $f(\boldsymbol{x})$ and the optimum. Under the model $f_t$, this (simple) regret manifests as a random variable

$$r_t(\boldsymbol{x}) = f_t^* - f_t(\boldsymbol{x}) \qquad\qquad f_t^* = \sup_{\boldsymbol{x} \in \mathcal{X}} f_t(\boldsymbol{x}). \quad (2)$$

Given a regret bound $\epsilon > 0$ and a risk tolerance $\delta > 0$, we would like to stop searching once we have found a point so that $r_t(\boldsymbol{x}) \leq \epsilon$ with probability at least $1 - \delta$ and refer to this stopping rule as a *probabilistic regret bound* (PRB). Probabilities of this sort are usually intractable and we will therefore estimate them via sampling. To this end, we denote the probability that a point $\boldsymbol{x}$ is

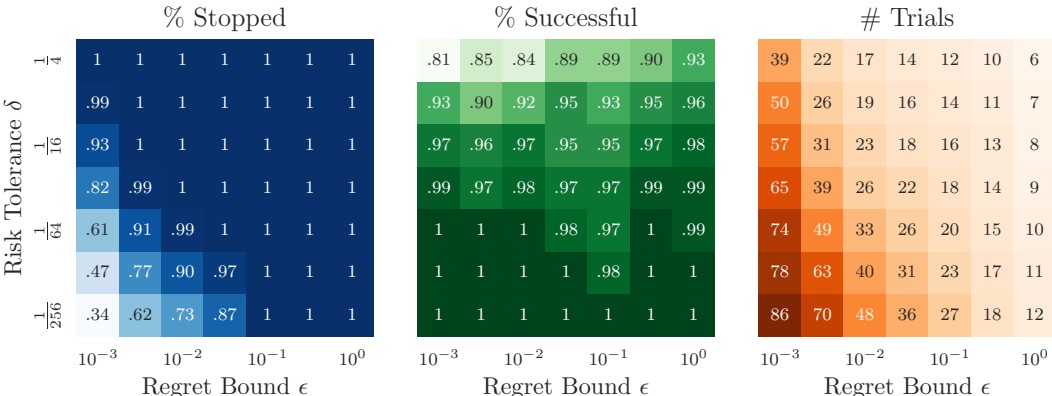

Figure 1: Overview of PRB stopping behavior when $f : [0, 1]^2 \to \mathbb{R}$ is drawn from a model with noise variance $\gamma^2 = 10^{-4}$. Regret bounds $\epsilon > 0$ dictate how close $f(\boldsymbol{x})$ must be to the optimum $f^*$ for $\boldsymbol{x} \in \mathcal{X}$ to be satisfactory. Tolerances $\delta > 0$ upper bound the chance of returning an unsatisfactory point. *Left:* Percent of runs that stopped before time $T = 128$. *Middle:* Percent of stopped runs that returned $\epsilon$-optimal points. *Right:* Median number of trials performed by stopped runs.

$\epsilon$-optimal and an associated Monte Carlo estimator by

$$\Psi_t(\boldsymbol{x}) = \mathbb{P}(r_t(\boldsymbol{x}) \leq \epsilon) \qquad \Psi_t^n(\boldsymbol{x}; \epsilon) = \frac{1}{n} \sum_{i=1}^n \mathbb{1}\big(r_t^i(\boldsymbol{x}) \leq \epsilon\big), \qquad (3)$$

where $r_t^i(\boldsymbol{x})$ is the $i$-th independent draw of the model-based regret (2). We will shortly explore how to construct estimators $\Psi_t^n(\boldsymbol{x})$ and use them to decide whether $\Psi_t(\boldsymbol{x})$ is above or below a level $\lambda \in \mathbb{R}$ in a manner that is both cost efficient and robust to estimation errors. First, however, let us introduce some basic terminology that will help us reason about potential failure modes.

We will say the estimator produces a *false positive* if $\Psi_t^n(\boldsymbol{x}) \geq \lambda > \Psi_t(\boldsymbol{x})$ and a *true positive* if $\Psi_t^n(\boldsymbol{x}) \geq \lambda \wedge \Psi_t(\boldsymbol{x}) \geq \lambda$. Since either scenario may lead to an unsatisfactory solution, the level $\lambda$ that we compare against must exceed $1 - \delta$. Accordingly, let $\delta_{\mathrm{mod}}$ and $\delta_{\mathrm{est}}$ be nonzero probabilities such that $\delta_{\mathrm{mod}} + \delta_{\mathrm{est}} \leq \delta$. By defining $\lambda = 1 - \delta_{\mathrm{mod}}$, we will use $\delta_{\mathrm{mod}}$ to limit the chance that a point $\boldsymbol{x}$ is not $\epsilon$-optimal even though $\Psi_t^n(\boldsymbol{x})$ produced a true positive. Conversely, we will use $\delta_{\mathrm{est}}$ to control the probability of encountering a false positive (see Section 3.2). This pattern guarantees that if $\Psi_t^n(\boldsymbol{x}) \geq \lambda$, then $\boldsymbol{x}$ is $\epsilon$-optimal with probability at least $1 - \delta$ under the model.

Algorithm 1 sketches a typical BO loop with the proposed stopping rule. At each iteration, we obtain a model for the data. We then select candidate solutions $\mathbf{C} \subseteq \mathcal{X}$ and estimate their probabilities of being $\epsilon$-optimal under the model. If an estimate is greater than $1 - \delta_{\mathrm{mod}}$, then the corresponding point satisfies the stopping conditions with probability at least $1 - \delta$ and we terminate; otherwise, we press on.

The rest of this section examines two key questions: how to simulate model-based regrets $r_t(\boldsymbol{x})$ when $|\mathcal{X}|$ is large (or infinite) and how to avoid false positives due to estimation error. Appendix A explores related topics such as how to choose $\mathbf{C}$ and schedule $\delta_{\mathrm{est}}^t$.

Figure 1 shows how the proposed algorithm behaves for different choices of $\epsilon$ and $\delta$. Data was generated

---

**Algorithm 1** BO with Monte Carlo PRB

1: **input** data $\mathcal{D} \in (\mathcal{X} \times \mathbb{R})^{T_0}$, limit $T \in \mathbb{N}$, and parameters $\epsilon, \delta_{\mathrm{mod}}, \delta_{\mathrm{est}} > 0$
2: $(\delta_{\mathrm{est}}^t) \leftarrow \mathtt{getSchedule}_\delta(\delta_{\mathrm{est}}, T - T_0)$
3: **for** $t = T_0, \ldots, T$ **do**
4:     $f \leftarrow \mathtt{getModel}(\mathcal{D})$
5:     $\mathbf{C} \leftarrow \mathtt{getCandidates}(f, \mathcal{D})$
6:     $\overline{\mathbf{Z}} \leftarrow \mathtt{MC\text{-}PRB}(\mathbf{C}; f, \epsilon, \delta_{\mathrm{mod}}, \delta_{\mathrm{est}}^t)$
7:     **if** $\max \overline{\mathbf{Z}} \geq 1 - \delta_{\mathrm{mod}}$ **then**
8:         **break**
9:     $\boldsymbol{x} \leftarrow \mathtt{getNextQuery}(f, \mathcal{D})$
10:     $\mathcal{D} \leftarrow \mathcal{D} \cup \{(\boldsymbol{x}, y(\boldsymbol{x}))\}$
11: **return** $\{\boldsymbol{x}_i \in \mathbf{C} : \overline{Z}_i = \max \overline{\mathbf{Z}}\}$

---

by running BO a hundred times and sampling $r_t(\boldsymbol{s}_t)$ a thousand times per step using the strategy from Section 3.1. Stopping decisions were then made by comparing estimators $\Psi_t^n(\boldsymbol{s}_t)$ with $\lambda = 1 - \delta_{\mathrm{mod}}$, where $\delta_{\mathrm{mod}} = \delta/2$. These results do not take advantage of the testing paradigm introduced in Section 3.2, but accurately reflects the algorithm's behavior. In particular, we see that the number of function evaluations performed by each run automatically adapts to the definition of $(\epsilon, \delta)$-optimality.

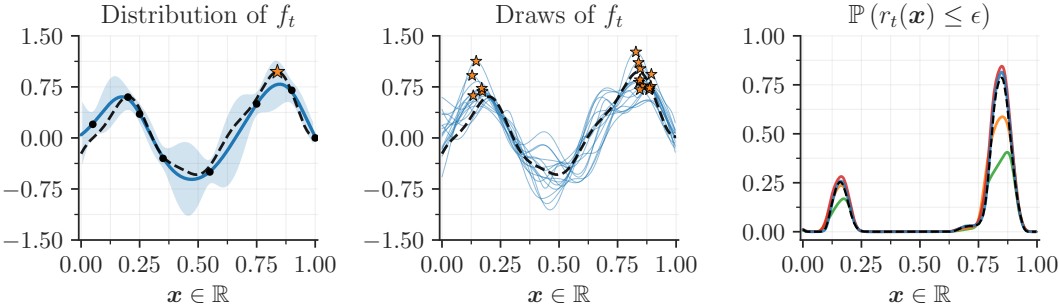

Figure 2: *Left:* Posterior mean and two standard deviations of $f$ (blue) given eight noisy observations (black dots). The goal is to find a point $\boldsymbol{x} \in \mathcal{X}$ whose true function (black) value is within $\epsilon > 0$ of the optimum $f^*$ (orange star). *Middle:* Draws of $f_t \sim \mathcal{GP}(\mu_t, k_t)$ and $f_t^*$ (orange stars). *Right:* Estimators for $\Psi_t$. Ground truth (dashed black) was established using location-scale sampling on a dense grid. The joint-sampling strategy from Section 3.1 is shown in blue. Competing methods analytically integrated out $f_t(\boldsymbol{x}) \mid f_t^*$ by approximating it with: $f_t(\boldsymbol{x})$, $f_t(\boldsymbol{x}) \mid f_t(\boldsymbol{x}) \le f_t^*$, or $f_t(\boldsymbol{x}) \mid f_t(\boldsymbol{x}) \le f_t^* \wedge f_t(\boldsymbol{x}_t^*) = f_t^*$ where $f_t^*$ and $\boldsymbol{x}_t^* \in \arg\max_{\boldsymbol{x} \in \mathcal{X}} f_t(\boldsymbol{x})$ were jointly sampled.

## 3.1 How to simulate stopping conditions

This section describes how to simulate whether a point $\boldsymbol{x} \in \mathcal{X}$ satisfies the chosen stopping conditions. For PRB, this amounts to sampling Bernoulli random variables $\mathbb{1}(r_t(\boldsymbol{x}) \le \epsilon)$. We propose to generate this term by maximizing draws of $f_t$. When dealing with parametric models, function draws are obtained by sampling parameter vectors. For GPs, analogous logic may be enacted by using a parametric approximation to the prior [52], as outlined below. This approximate sampling step is necessary because the time complexity for exactly simulating $f_t(\mathbf{X})$ scales cubically in $|\mathbf{X}|$.

Let $\boldsymbol{\phi} : \mathcal{X} \to \mathbb{R}^m$ be a finite-dimensional feature map so that, $\forall \boldsymbol{x}, \boldsymbol{x}' \in \mathcal{X}$, $\boldsymbol{\phi}(\boldsymbol{x})^\top \boldsymbol{\phi}(\boldsymbol{x}') \approx k(\boldsymbol{x}, \boldsymbol{x}')$. Note that feature maps of this sort are readily available for many popular covariance functions [37, 52]. Equipped with such a map, we may approximate a prior $f \sim \mathcal{GP}(0, k)$ with a Bayesian linear model

$$\hat{f}(\cdot) = \boldsymbol{\phi}(\cdot)^\top \boldsymbol{w} \qquad\qquad \boldsymbol{w} \sim \mathcal{N}(\mathbf{0}, \mathbf{I}). \tag{4}$$

Letting $\boldsymbol{\Lambda} = k(\mathbf{X}_t, \mathbf{X}_t) + \gamma^2 \mathbf{I}$ and $\boldsymbol{\varepsilon} \sim \mathcal{N}(\mathbf{0}, \gamma^2 \mathbf{I})$, this linear model may be used to generate draws from an approximate posterior by sampling $\boldsymbol{w}$ from the prior and using Matheron's rule to write [51]

$$f_t(\cdot) \stackrel{d}{\approx} \hat{f}(\cdot) + k(\cdot, \mathbf{X}_t)\boldsymbol{\Lambda}^{-1}\big[\boldsymbol{y}_t - \hat{f}(\mathbf{X}_t) - \boldsymbol{\varepsilon}\big]. \tag{5}$$

For each draw of $f_t$, the remaining problem is now to evaluate $\mathbb{1}(r_t(\boldsymbol{x}) \le \epsilon)$. We suggest using multi-start gradient ascent. In our case, we performed an initial random search to identify promising starting locations and then used a quasi-Newton method [28] to optimize. A helpful insight is that we do not need to find $f_t^*$ per se. Rather, it suffices to determine whether there exists a point $\boldsymbol{x}' \in \mathcal{X}$ such that $f_t(\boldsymbol{x}') - f_t(\boldsymbol{x}) > \epsilon$. This property can be exploited to accelerate simulating $\mathbb{1}(r_t(\boldsymbol{x}) \le \epsilon)$; however, its benefits wane as $\Psi_t(\boldsymbol{x})$ increases because $r_t(\boldsymbol{x}) \le \epsilon$ implies that no such point $\boldsymbol{x}'$ exists.

The right panel of Figure 2 compares different estimators for $\Psi_t$. For simplicity, assume that $f_t$ is sample continuous so that it almost surely attains its supremum on $\mathcal{X}$. The goal of this plot is to highlight challenges inherent to conditioning on the maximum. We not only need to upper bound $f_t$, but also account for the point(s) at which the maximum is achieved. This explains why the red estimator $\mathbb{E}_{\boldsymbol{x}_t^*, f_t^*}[\mathbb{P}(f_t^* - f_t(\boldsymbol{x}) \le \epsilon \mid f_t(\boldsymbol{x}_t^*) = f_t^*, f_t(\boldsymbol{x}) \le f_t^*)]$ outperforms the orange one $\mathbb{E}_{f_t^*}[\mathbb{P}(f_t^* - f_t(\boldsymbol{x}) \le \epsilon)]$, while the green one $\mathbb{E}_{f_t^*}[\mathbb{P}(f_t^* - f_t(\boldsymbol{x}) \le \epsilon \mid f_t(\boldsymbol{x}) \le f_t^*)]$ fails to do so. We opted to avoid these issues by sampling $f_t(\boldsymbol{x})$ jointly with $f_t^*$ rather than marginalizing it out. The resulting blue estimator is seen to more accurately follow the gold standard shown in black.

Lastly, it should be said that the suggested sampling procedure introduces a yet-to-be-determined amount of error in practice, since draws of $f_t$ are not only approximate but non-convex. Initial results suggest these errors are small (see Figure 2), however we leave this as a topic for future investigation.

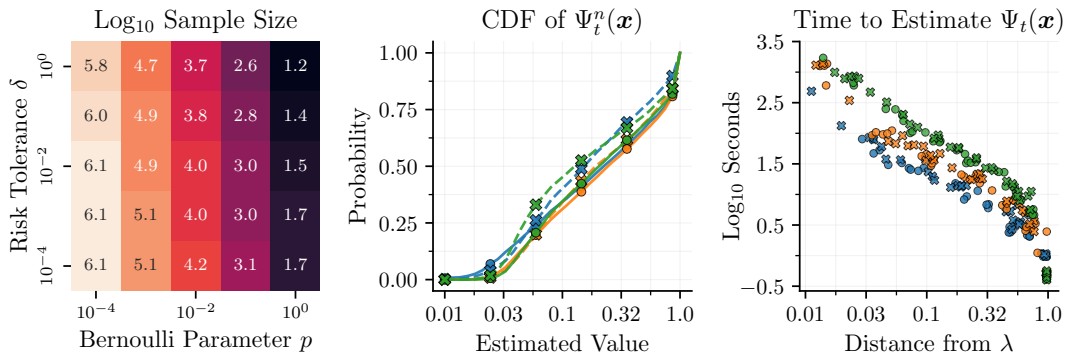

Figure 3: *Left:* Median number of draws used by Algorithm 2 to decide if the expectation of a Bernoulli random variable $Z \sim \text{Bern}(p)$ exceeds $\lambda = 10^{-5}$ (chosen arbitrarily). *Middle:* Empirical CDFs of $\Psi_t^n$ when optimizing draws from known priors $\mathcal{GP}(0, k)$ in two, four, and six dimensions with noise variance $\gamma^2 = 10^{-6}$ (solid, $\circ$) or $\gamma^2 = 10^{-2}$ (dashed, $\times$). PRB parameters were set to $\epsilon = 0.1$ and $\delta_{\text{mod}} = \delta_{\text{est}} = 2.5\%$. *Right:* Runtimes for Algorithm 2 using the generative strategy from Section 3.1 and a (wall) time limit of roughly one thousand seconds.

## 3.2 How to efficiently make robust decisions with Monte Carlo estimates

This section discusses the general problem of using samples to decide whether the expectation of a random variable $Z \in \{0, 1\}$ exceeds a level $\lambda \in [0, 1]$. For PRB, $\lambda = 1 - \delta_{\text{mod}}$ and each evaluation of the stopping rule corresponds to a unique $Z = \mathbb{1}(r_t(\boldsymbol{x}) \leq \epsilon)$. We will show how to make probably-correct decisions using a minimal number of samples $n \in \mathbb{N}$. In doing so, we first discuss confidence intervals for $\mathbb{E}[Z]$ based on a collection of i.i.d. draws $\mathbf{Z}_n = \{z_i : i = 1 \ldots, n\}$.

There are many techniques for generating intervals that contain $\mathbb{E}[Z]$ with (coverage) probability at least $1 - \delta_{\text{est}}$. Clopper & Pearson [13] gave an exact recipe for constructing confidence intervals for Bernoulli random variables $Z$ as

$$\text{CI}(\mathbf{Z}_n; \delta) = \left[ B\left(\tfrac{\delta}{2}; k, n - k + 1\right), B\left(1 - \tfrac{\delta}{2}; k + 1, n - k\right) \right], \tag{6}$$

where $B$ denotes the beta quantile function and $k = \sum_{i=1}^n z_i$ is the number of successes in $n$ draws. It is also possible to take a Bayesian by placing a prior on $\mathbb{E}[Z]$. Differences between this Bayesian approach and (6) were observed to be minimal however, so we opted to avoid modeling $\mathbb{E}[Z]$. For further discussion, see Appendix A.1.

Given an estimate $\overline{Z}_n = \frac{1}{n} \sum_{i=1}^n z_i$ and a confidence interval $\mathcal{I}_n = \text{CI}(\mathbf{Z}_n; \delta_{\text{est}})$, it follows that

$$\mathbb{E}[Z] \in \mathcal{I}_n \text{ and } \lambda \notin \mathcal{I}_n \implies \underbrace{\mathbb{1}\left(\mathbb{E}[Z] \geq \lambda\right)}_{\text{ground truth}} = \underbrace{\mathbb{1}\left(\overline{Z}_n \geq \lambda\right)}_{\text{decision}}. \tag{7}$$

If $\mathcal{I}_n$ collapses to a point as $n \to \infty$, then there exist sample sizes such that $\lambda \notin \mathcal{I}_n$, whereupon the conclusion from (7) holds with probability at least $1 - \delta_{\text{est}}$.[1] Said differently, we can lower bound the probability that we correctly decide whether $\mathbb{E}[Z] \geq \lambda$ by generating enough draws of $Z$. With these details in mind, we now review an algorithm for adaptively choosing $n \in \mathbb{N}$ in order to make probably-correct decisions using as few samples as possible—which is crucial when simulating $Z$ is computationally intensive as in Section 3.1.

The general idea of Algorithm 2 is to perform a series of tests (each using more samples than the last), until a confidence interval for $\mathbb{E}[Z]$ is narrow enough for a decision to be made. To better understand this, start by defining two sequences: sample sizes $(n_j)$ and risk tolerances $(d_j)$. The sizes should be increasing, while the tolerances should be positive and satisfy $\sum_{j=1}^\infty d_j \leq \delta_{\text{est}}$.

Next, imagine that we generate draws of $Z$ in batches of size $n_j - n_{j-1}$, where $n_0 = 0$. At each round of sampling $j$, we construct an interval $\mathcal{I}_{n_j}$ that contains $\mathbb{E}[Z]$ with probability at least $1 - d_j$. If $\lambda \notin \mathcal{I}_{n_j}$, we use $\overline{Z}_{n_j}$ to decide whether $\mathbb{E}[Z] \geq \lambda$. Otherwise, we proceed to the next iteration.

---

[1] We may that assume $\mathbb{P}(\lambda = \mathbb{E}[Z]) = 0$. Here, $\mathbb{E}[Z] = \Psi_t(\boldsymbol{x})$ is only random prior to observing $y(\mathbf{X}_t)$.

Per (7), this algorithm only makes an incorrect decision if the final interval fails to contain $\mathbb{E}[Z]$. By definition of $(d_j)$ and the union bound however, the chance of any interval not containing $\mathbb{E}[Z]$ is at most $\delta_{\text{est}}$. Hence, the algorithm makes the correct decision with probability at least $1 - \delta_{\text{est}}$.

Algorithm 2 was inspired by bandit methods, such as Mnih et al. [33] and references contained therein, who previously studied how concentration inequalities can be used to iteratively test whether $\mathbb{P}(|\overline{Z}_n - \mathbb{E}[Z]| \leq \epsilon \mathbb{E}[Z]) \geq 1 - \delta_{\text{est}}$. Algorithm 2 is closer to Bardenet et al. [6] however, who used a similar strategy to decide whether to accept Metropolis-Hastings proposals based on subsampled estimates of the data log-likelihood.

---

**Algorithm 2** Monte Carlo PRB

1: **input** point $\boldsymbol{x} \in \mathcal{X}$, model $f \colon \Omega \times \mathcal{X} \to \mathbb{R}$
 and parameters $\epsilon, \delta_{\text{mod}}, \delta_{\text{est}} > 0$
2: $(d_j) \leftarrow \texttt{getSchedule}_d(\delta_{\text{est}})$
3: $(n_j) \leftarrow \texttt{getSchedule}_n()$
4: $\mathbf{Z} \leftarrow \emptyset$
5: **for** $j = 1, 2, \ldots$ **do**
6:    **while** $|\mathbf{Z}| < n_j$ **do**
7:       $f^i \leftarrow \texttt{getSample}(f)$
8:       $f_*^i \leftarrow \sup_{\boldsymbol{x} \in \mathcal{X}} f^i(\boldsymbol{x})$
9:       $\mathbf{Z} \leftarrow \mathbf{Z} \cup \{ \mathbb{1}(f_*^i - f^i(\boldsymbol{x}) \leq \epsilon) \}$
10:    **if** $1 - \delta_{\text{mod}} \notin \text{CI}(\mathbf{Z}, d_j)$ **then**
11:       **break**
12: **return** $\texttt{Average}(\mathbf{Z})$

---

Extending our earlier argument, let $(\delta_{\text{est}}^t)$ be a sequence of risk tolerances such that $\sum_{t=1}^{\infty} \delta_{\text{est}}^t \leq \delta_{\text{est}}$. If Algorithm 2 is run at each BO step $t$ with schedule $(d_j^t)$ such that $\sum_{j=1}^{\infty} d_j^t \leq \delta_{\text{est}}^t$, then the chance of encountering a false positive at any step is bounded from above by $\delta_{\text{est}}$. Consequently, the decision to stop will be correct with probability at least $1 - \delta_{\text{est}}$.

We followed Mnih et al. [33] by defining $d_j^t = j^{-\alpha} \frac{(\alpha-1)}{\alpha} \delta_{\text{est}}^t$ and $n_j = \lceil \beta^{j-1} N \rceil$. We set $\alpha = 1.1$ so that $(d_j^t)$ decayed slowly, $\beta = 1.5$ such that $(n_j)$ grew reasonably quickly, and $N = 64$ because smaller starting values took longer to run. These choices impact the algorithm's runtime, not its validity. Using a geometric schedule for $(n_j)$ prevents $(d_j^t)$ from rapidly shrinking due to a large number of tests being performed with very few samples. In exchange, this schedule can lead to nearly $\beta$ times too many samples being requested.

The left panel of Figure 3 shows how many samples Algorithm 2 used to decide whether $\mathbb{E}[Z] \geq \lambda$ for $Z \sim \text{Bern}(p)$. As $p \to 1$, the distance between $\overline{Z}_n$ and $\lambda$ tends to increase and decisions can be made with wider confidence intervals constructed using fewer draws. As $\delta_{\text{est}} \to 1$, these intervals shrink and decisions can similarly be made using fewer samples. The middle panel visualizes the empirical CDF of estimates $\Psi_t^n$ from BO experiments described in Section 5. For most of a typical BO run's life cycle, these estimates are far from $\lambda = 1 - \delta_{\text{mod}}$ so decisions can be made efficiently. This pattern is reflected in the rightmost panel, which illustrates the savings provided by Algorithm 2.

## 4 Analysis

We show that Bayesian optimization with the PRB stopping rule terminates under mild assumptions. Further, we prove that the given algorithm is correct in the sense that it returns an $(\epsilon, \delta)$-optimal point under the model. We begin by discussing the assumptions made throughout this section, which are:

    **A1.** The search space $\mathcal{X} = [0, 1]^D$ is a unit hypercube.

    **A2.** There exists a constant $L_k > 0$ so that, $\forall \boldsymbol{x}, \boldsymbol{x}' \in \mathcal{X}, |k(\boldsymbol{x}, \boldsymbol{x}) - k(\boldsymbol{x}, \boldsymbol{x}')| \leq L_k \|\boldsymbol{x} - \boldsymbol{x}'\|_{\infty}$.

    **A3.** The sequence of query locations $(\boldsymbol{x}_t)$ is almost surely dense in $\mathcal{X}$.

A1 and A2 guarantee it is possible for the maximum posterior variance to become arbitrarily small given a finite number of observations. Note that if hyperparameters change over time, we only require that the (best) Lipschitz constant $L_k$ and noise variance $\gamma^2$ do not grow without bounds as $t \to \infty$. Combined with these assumptions, A3 implies that, for any $C > 0$, there exists a time $T \in \mathbb{N}_0$ such that, $\forall t \geq T, \max_{\boldsymbol{x} \in \mathcal{X}} k_t(\boldsymbol{x}, \boldsymbol{x}) \leq C$ with probability one. More generally, A3 is necessary to ensure convergence when all we known is that $\mathcal{X}$ is compact and $f$ is continuous [45].

When A1 and A2 hold, popular strategies often produce almost surely dense sequences $(\boldsymbol{x}_t)$. For instance, Vazquez & Bect [47] proved Probability of Improvement [25] and Expected Improvement [38] exhibit this behavior for many covariance functions $k$ when $f$ is directly observed. In Appendix B, we show that this result holds for continuous acquisition functions that value informative queries over unambiguous ones. This family includes well-known acquisition functions such as Knowledge Gradient [15], Entropy Search [20], and variants thereof [21, 49]. Finally, dense sequences can be

guaranteed by introducing a small chance for queries to be selected at random from an appropriately chosen distribution [44].

We first prove that points which maximize the posterior mean eventually satisfy the PRB criterion and then use this result to demonstrate convergence and correctness.

**Proposition 1.** *Under assumptions A1–A3 and for all regret bounds $\epsilon > 0$ and risk tolerances $\delta > 0$, there almost surely exists $T \in \mathbb{N}_0$ so that, at each time $t \geq T$, every $\boldsymbol{s}_t \in \arg\max_{\boldsymbol{x} \in \mathcal{X}} \mu_t(\boldsymbol{x})$ satisfies*

$$\Psi_t(\boldsymbol{x}; \epsilon) = \mathbb{P}(r_t(\boldsymbol{s}_t) \leq \epsilon) \geq 1 - \delta. \tag{8}$$

*Sketch.* We sketch the proof below and provide full details in Appendix B for details. Consider the centered process $g_t(\cdot) = [f_t(\cdot) - f_t(\boldsymbol{s}_t)] + [\mu_t(\boldsymbol{s}_t) - \mu_t(\cdot)]$. Since the second term is nonnegative, $g_t^* = \sup_{\boldsymbol{x} \in \mathcal{X}} g_t(\boldsymbol{x}) \geq r_t(\boldsymbol{s}_t) = f_t^* - f(\boldsymbol{s}_t)$ and it suffices to upper bound the probability that $g_t^* \geq \epsilon$. For $\epsilon > \mathbb{E}(g_t^*)$, such a bound may be constructed by using the Borell-TIS inequality [8, 46] to write

$$\mathbb{P}(g_t^* \geq \epsilon) \leq \exp\left(-\frac{1}{2}\left[\frac{\epsilon - \mathbb{E}(g_t^*)}{\sigma_t}\right]^2\right) \qquad\qquad \sigma_t^2 = \max_{\boldsymbol{x} \in \mathcal{X}} \mathrm{Var}[g_t(\boldsymbol{x})]. \tag{9}$$

Since $\mathbb{E}(g_t^*)$ and (9) both vanish as $\sigma_t$ decreases, the claim holds so long as $\lim_{t \to \infty} \sigma_t = 0$. □

Similar ideas can be found in Grünewälder et al. [18], who proved that the expected supremums of centered process like $g_t$ go to zero as $(\boldsymbol{x}_t)$ becomes increasingly dense in $\mathcal{X}$. In Appendix B, we extend this result to the setting where observations are corrupted by i.i.d. Gaussian noise and combine it with the Borell-TIS inequality to show the probability that $r_t(\boldsymbol{s}_t) \geq \epsilon$ vanishes. We also give a simple corollary for the case where solutions $\boldsymbol{s}_t$ belong to $\mathbf{X}_t$. Next, we show that BO not only stops when Algorithm 2 is used to evaluate the proposed rule but does so correctly.

**Proposition 2.** *Suppose assumptions A1–A3 hold. Given a risk tolerance $\delta > 0$, define nonzero probabilities $\delta_{\mathrm{mod}}$ and $\delta_{\mathrm{est}}$ such that $\delta_{\mathrm{mod}} + \delta_{\mathrm{est}} \leq \delta$ and let $(\delta_{\mathrm{est}}^t)$ be a positive sequence so that $\sum_{t=0}^{\infty} \delta_{\mathrm{est}}^t \leq \delta_{\mathrm{est}}$. For any regret bound $\epsilon > 0$, if Algorithm 2 is run at each step $t \in \mathbb{N}_0$ with tolerance $\delta_{\mathrm{est}}^t$ to decide whether a point $\boldsymbol{s}_t \in \arg\max_{\boldsymbol{x} \in \mathcal{X}} \mu_t(\boldsymbol{x})$ satisfies the stopping criterion*

$$\Psi_t(\boldsymbol{x}; \epsilon) = \mathbb{P}(r_t(\boldsymbol{s}_t) \leq \epsilon) \geq 1 - \delta_{\mathrm{mod}}, \tag{10}$$

*then BO almost surely terminates and returns an $(\epsilon, \delta)$-optimal solution under the model.*

*Proof.* By Proposition 1, there almost surely exists an $S \in \mathbb{N}_0$ so that $t \geq S \implies \Psi_t(\boldsymbol{s}_t) \geq 1 - \delta_{\mathrm{mod}}$. Further, because $\Psi_t^n(\boldsymbol{s}_t)$ is unbiased, there exist times $t \geq T$ at which Algorithm 2 produces true positives $\Psi_t^n(\boldsymbol{s}_t) \geq 1 - \delta_{\mathrm{mod}} \wedge \Psi_t(\boldsymbol{s}_t) \geq 1 - \delta_{\mathrm{mod}}$. Hence, BO stops with probability one. If BO terminates at time $T \in \mathbb{N}_0$, then the probability that $\boldsymbol{s}_T$ is not $\epsilon$-optimal is less than or equal to $\delta_{\mathrm{mod}}$ in the event of a true positive and one otherwise. Since false positives $\Psi_t^n(\boldsymbol{s}_t) \geq 1 - \delta_{\mathrm{mod}} > \Psi_t(\boldsymbol{s}_t)$ occur with probability at most $\delta_{\mathrm{est}}$, it follows that $\boldsymbol{s}_T$ is $\epsilon$-optimal with probability at least $1 - \delta$. □

In summary, we can design statistical tests to mitigate the risk of premature stopping due to random fluctuations in Monte Carlo estimators like $\Psi_t^n$. Moreover, we can schedule these tests to ensure that points which pass them are sufficiently likely (under the model) to satisfy our stopping conditions. If the model is correct, we can therefore guarantee that a satisfactory solution is returned with high probability. Provided that one or more points almost surely satisfy the rule as $t \to \infty$, this result holds if we can simulate whether solutions are satisfactory and bound the error in the resulting estimator.

## 5 Experiments

To shed light on how our algorithm behaves in practice, we conducted a series of experiments. Focal questions here included: i) how does PRB perform in comparison to existing stopping rules, ii) how do these rules respond to different types of problems, and iii) what is the impact of model mismatch.

Experiments were performed by first running BO with conservatively chosen budgets $T \in \mathbb{N}$. We then stepped through each saved run with different stopping rules to establish stopping times and terminal performance. This paradigm ensured fair comparisons and reduced compute overheads. We performed a hundred independent BO runs for all problems other than hyperparameter tuning for convolutional neural networks (CNNs) on MNIST [14], where only fifty runs were carried out.

| Problem | $D$ | $T$ | Oracle$^\dagger$ | Budget$^\dagger$ | Acq | $\Delta$CB | $\Delta$ES | PRB (ours) |
|---|---|---|---|---|---|---|---|---|
| **GP$^\dagger$** $10^{-6}$ | 2 | 64 | 10 (100) | 17 (96) | 28 (100) | **16 (96)** | 22 (99) | 17 (97) |
| **GP$^\dagger$** $10^{-2}$ | 2 | 128 | 11 (100) | 22 (96) | 78 (100) | 128 (100) | 54 (100) | **23 (99)** |
| **GP$^\dagger$** $10^{-6}$ | 4 | 128 | 27 (100) | 64 (95) | 90 (100) | **51 (97)** | 93 (100) | 64 (99) |
| **GP$^\dagger$** $10^{-2}$ | 4 | 256 | 30 (100) | 94 (95) | 106 (98) | 256 (100) | 144 (97) | **86 (96)** |
| **GP$^\dagger$** $10^{-6}$ | 6 | 256 | 40 (99) | 124 (95) | 142 (98) | 150 (98) | 256 (99) | **134 (98)** |
| **GP$^\dagger$** $10^{-2}$ | 6 | 512 | 65 (100) | 227 (96) | **181 (96)** | 512 (100) | 278 (99) | 235 (100) |
| **GP** $10^{-6}$ | 4 | 128 | 35 (100) | 79 (95) | **92 (100)** | 41 (66) | 77 (94) | 61 (88) |
| **GP** $10^{-2}$ | 4 | 256 | 51 (100) | 157 (95) | **128 (97)** | 256 (100) | 160 (96) | 100 (92) |
| **Branin** | 2 | 128 | 19 (100) | 25 (95) | 64 (100) | 36 (100) | 38 (100) | **33 (99)** |
| **Hartmann** | 3 | 64 | 14 (100) | 22 (96) | 26 (100) | 18 (90) | 21 (97) | **19 (100)** |
| **Hartmann** | 6 | 64 | 36 (67) | 256 (67) | 40 (67) | 38 (67) | 62 (67) | 40 (64) |
| **Rosenbrock** | 4 | 96 | 34 (100) | 46 (95) | 95 (100) | 88 (100) | 98 (100) | **84 (100)** |
| **CNN** | 4 | 256 | 5 (100) | 11 (96) | 64 (100) | 64 (100) | 64 (100) | **17 (100)** |
| **XGBoost** | 3 | 128 | 4 (100) | 8 (97) | 128 (100) | 90 (100) | 51 (100) | **28 (99)** |

Table 1: Median stopping times and success rates when seeking $(\epsilon, \delta)$-optimal points on $\mathcal{X} = [0, 1]^D$ given an upper limit of $T \in \mathbb{N}$ function evaluations. For GP objectives, number beside each name specify noise levels $\gamma^2$. Superscripts $^\dagger$ indicate that model or stopping rule parameters were given by an oracle. For each problem, non-oracle methods that returned $\epsilon$-optimal points at least $1 - \delta$ percent of the time using the fewest function evaluations are shown in **blue**.

Despite the general notation of the paper, all problems were defined as minimization tasks. Additional details and results can be found in Appendices C and D, respectively; and, code is available online at https://github.com/j-wilson/trieste_stopping.

Each BO run was tasked with finding an $\epsilon$-optimal point with probability at least $1 - \delta = 95\%$. On the Rosenbrock-4 fine-tuning problem, we used a regret bound $\epsilon = 10^{-4}$. For CNNs, we aimed to be within $\epsilon = 0.5\%$ of the best test error (i.e., misclassification rate) seen across all runs, namely $0.62\%$. Likewise, when fitting XGBoost classifiers [12] for income prediction [7], we sought to be within $1\%$ of the best found test error of $12.89\%$. For all other problems, we set $\epsilon = 0.1$.

For PRB, we divided $\delta$ evenly between $\delta_{\text{est}}$ and $\delta_{\text{mod}}$. Since experiments were carried out using preexisting BO runs that each began with five random trials and ended at times $T$, we employed a constant schedule $\delta_{\text{est}}^t = \frac{1}{T-5}\delta_{\text{est}}$ for risk tolerances at steps $t \in \mathbb{N}_0$. Parameter schedules for Algorithm 2 are discussed in Section 3.2.

As a practical concession, we limited each run of Algorithm 2 to a thousand draws of $f_t$ and used the resulting estimate to decide whether to stop—even if the corresponding confidence interval was not narrow enough to afford guarantees. Results under this setup were consistent with preliminary experiments in which Algorithm 2 was run using a fifteen minute time limit. Finally, when optimizing draws from GP priors in six dimensions with noise $\gamma^2 = 10^{-2}$, we evaluated PRB once every five steps to expedite these experiments.

## 5.1 Baselines

We tested several baselines, some of which were granted access to information that would usually be unavailable (indicated by a dagger $\dagger$). We summarize these as follows[2]:

**B1.** Oracle$^\dagger$: stops once an $\epsilon$-optimal point has been evaluated.

**B2.** Budget$^\dagger$: stops after a fixed number of trials chosen by an oracle for each problem.

**B3.** Acq [23, 35]: stops when the acquisition value of the next query is negligible.

---

[2]Note that these descriptions do not account for the presence of a link function (see Appendix C.2).

**B4.** $\Delta$CB [29]: stops once the gap between confidence bounds is less-equal to $C > 0$, i.e.

$$\max_{\boldsymbol{x} \in \mathcal{X}} \mathrm{UCB}_t(\boldsymbol{x}) - \max_{\boldsymbol{x}' \in \mathbf{X}_t} \mathrm{LCB}_t(\boldsymbol{x}') \leq C \quad [\mathrm{U/L}]\mathrm{CB}_t(\boldsymbol{x}) = \mu_t(\boldsymbol{x}) \pm \sqrt{\beta_t k_t(\boldsymbol{x}, \boldsymbol{x})}$$

**B5.** $\Delta$ES [22]: stops when an upper bound on $\left| \mathbb{E}(f_t^*) - \mathbb{E}(f_{t-1}^*) \right|$ drops below a level.

B1 is the optimal stopping rule, but requires perfect information for $f$. Likewise, B2 is the optimal fixed budget for each problem. These budgets were defined post-hoc as the minimum number of trials such that at least $95\%$ percent of runs returned $\epsilon$-optimal points (where possible).

The remaining methods are all model-based and stop when target quantities are sufficiently small. For the chosen acquisition function (see Appendix C.3), B3 can be interpreted as the expected improvement in solution quality given an additional trial, i.e. $\mathbb{E}_{y_{t+1}} \left[ \max_{\boldsymbol{x} \in \mathbf{X}_{t+1}} \mu_{t+1}(\boldsymbol{x}) - \mu_{t+1}(\boldsymbol{s}_t) \right]$. Unfortunately, neither this quantity nor the change in the expected supremum used by B5 lend themselves to interpretation in terms of $(\epsilon, \delta)$-optimality. B4 does admit such an interpretation for appropriate choice of constant $\beta_t$ [43]; however, these constants are often difficult to obtain in practice however, so we followed [29] by defining $\beta_t = \frac{2}{5} \log\left(Dt^2\pi^2/6\delta\right)$.

To combat these issues, we gave baseline methods a competitive advantage by retroactively assigning cutoff values to ensure they achieved the desired success rate when optimizing draws from the model (denoted GP$^\dagger$). Specifically, cutoff values for B3–B5 were obtained by dividing regret bounds $\epsilon$ by the smallest powers of two for which this condition held—explicitly: $2^{15}$, $2^3$, and $2^4$ (respectively). Note that, in the absence of this fine tuning, these methods either proved unreliable or failed to stop within the allotted time depending on whether thresholds were too large or too small. For completion, additional results using $\epsilon$ as the cutoff value for B4 and B5 are presented in Appendix D.

The main results of this section are shown in Table 1 and key findings are discussed below.

## 5.2 Results with true models

When optimizing functions drawn from known GP priors, denoted GP$^\dagger$, the proposed stopping rule performed exactly as advertised and consistently returned $(\epsilon, \delta)$-optimal solutions. Moreover, PRB often requiring the fewest function evaluations. This result is not surprising when comparing with methods like B4 because an unbiased estimate to $\mathbb{P}(r_t(\boldsymbol{x}) \leq \epsilon)$ should exceed a level faster than a corresponding lower bound. In many cases, PRB achieved a higher success rate than the fixed budget oracle using a comparable or smaller number of trials. These gains occur because model-based stopping is able to exploit patterns in the data collected by individual runs.

Elsewhere, we observe that B4 struggled to terminate when faced with moderate noise levels $\gamma^2 = 10^{-2}$. This pathology likely emerges because, similar to alternative estimators discussed in Section 3.1, the method does not fully account for dependencies between $f_t^*$ and $f_t(\boldsymbol{x})$. As an extreme example, B4 may fail to terminate when a point $\boldsymbol{x} \in \mathbf{X}_t$ simultaneously maximizes upper and lower confidence bounds, despite the fact that $r_t(\boldsymbol{x} \mid \boldsymbol{x}_t^* = \boldsymbol{x}) = 0$.

Not surprisingly, BO runs that took longer to query an $\epsilon$-optimal point took longer to stop. However, the correlation between these terms paled in comparison to that of stopping times and $\alpha$-quantiles of regrets incurred by uniform random points (approximately, $0.35$ vs. $-0.75$). Said differently, PRB stopped faster when $f^*$ was an outlier. This pattern suggests that the one-step optimal strategy from Appendix C.3 is better at finding optimal solutions than verifying them. Future works may therefore wish to pursue stopping-aware approaches along the lines of McLeod et al. [32] or Cai et al. [11].

## 5.3 Results with maximum a posteriori models

In the real world, the high-level assumptions that govern how the model behaves (i.e., its hyperparameters) are tuned online as additional data is collected using Type-II maximum likelihood. We are therefore interested in seeing how discrepancies between the model and reality influence stopping behavior.

Results here were similar to the synthetic setting, albeit with some blemishes. Interestingly, the most glaring example of the risks posed by model mismatch occurred on the popular Hartmann-6 test function. Here, $33\%$ of BO runs overestimated the objective function's smoothness and converged to a local minimum of $-3.20$ rather than the global minimum of $-3.32$. It is worth noting, however,

that if $\epsilon = 0.1$ had been slightly larger, all stopping rules would have succeeded in at least $95\%$ of cases (see Appendix D). Along similar lines, models occasionally underestimated the kernel variance when optimizing draws from GP priors and stopped prematurely.

These results also indicate that both hyperparameter tuning problems (CNN and XGBoost) were fairly easy and this may have masked potential failure modes. The fixed budget oracle's performance demonstrates that there was still room for model-based stopping rules to fail, but we nevertheless recommend that these results be taken with a grain of salt.

In additional experiments, we indeed found that it was easy to construct cases where poor model fits led to poor stopping behavior. This vulnerability was large due to our choice of hyperpriors (see Appendix C.1), which were purposefully broad and uninformative. Overall, we argue that these results are both highly encouraging and also highlight the importance of uncertainty calibration. Potential remedies for this issue are discussed below.

Based on these findings, we suggest that model-based stopping be used with more conservative priors that, e.g., favor smaller lengthscales and larger variances. Alternatively, calibration issues may be alleviated by marginalizing over hyperparameters [41] or utilizing more expressive models. These options help reduce the risk of overly confident models leading to premature stopping. Along the same lines, we recommend using a large fixed budget as an auxiliary stopping rule to avoid cases where poor model fits cause the algorithm to converge very slowly (see Appendix A.3 for discussion).

## 6   Conclusion

To the best of our knowledge, results presented here are among the first of their kind for Bayesian optimization. We have given a practical algorithm for verifying whether a set of stopping conditions holds with high probability under the model. For the proposed stopping rule, we have further shown that the algorithm correctly terminates under mild technical conditions. If data is generated according to the model, we can therefore guarantee that BO is likely to return a satisfactory solution.

The methods we have shared are largely generic. Echoing the introduction, if you can simulate it then you can use it for stopping. While this approach is not without limitations, we believe that it will ultimately allow others to design stopping rules as they see fit. To the extent that it does, model-based stopping may one day become as common place as model-based optimization.

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

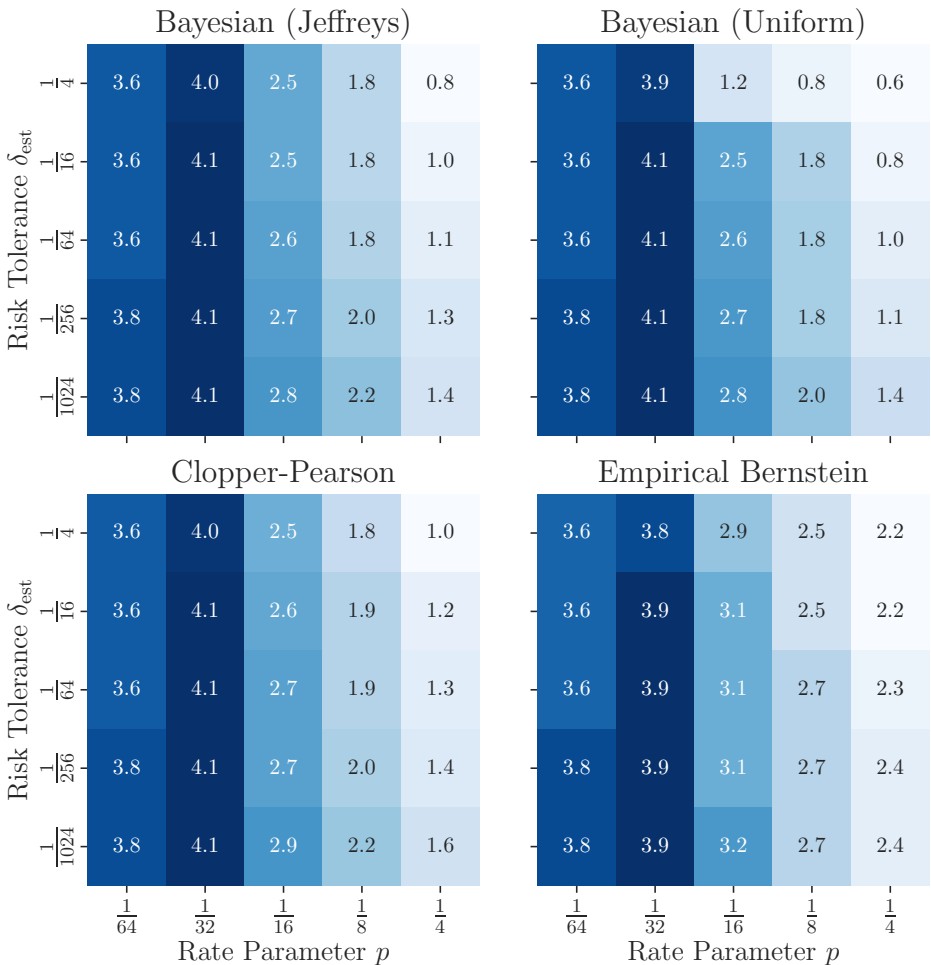

Figure 4: Median number of samples (show in $\log_{10}$) used by Algorithm 2 to decide if the expectation of $Z \sim \text{Bernoulli}(p)$ exceeds $\lambda = 2.5\%$ using different types of intervals with nominal coverage probability $1 - \delta_{\text{est}}$. The number of samples drawn is seen decrease in both $\delta_{\text{est}}$ and $|p - \lambda|$.

## A  Practical recommendations

This section aims to fill in some of the gaps left by Section 3 by providing further details for various subproblems and design choices encountered in practice.

### A.1  How to construct confidence intervals

As discussed in the text, given a random variable $Z \in \mathbb{R}$, there are different ways of generating intervals $\mathcal{I}_n \subseteq \mathbb{R}$ that contain the true parameter $\mathbb{E}[Z]$ with (nominal) coverage probability $1 - \delta_{\text{est}}$.

For Bernoulli random variables, Clopper-Pearson intervals (6) are a classic approach to this problem. This method is said to be "exact" because (instead of relying on the central limit theorem) it uses the fact that $X = \sum_{i=1}^{n} Z_i$ follows a Binomial distribution, where $Z_i$ is the $i$-th independent copy of $Z$. This method is also conservative: its (true) coverage probability is greater than or equal to $1 - \delta_{\text{est}}$.

Alternatively, one can take a Bayesian approach by placing a prior on success rate $p \in [0, 1]$ of the Binomial random variable $X \sim \text{Bin}(n, p)$. If this prior is chosen to be a Beta distribution $p \sim \text{Beta}(\alpha, \beta)$, then the posterior is conjugate and we have $p \mid X \sim \text{Beta}(\alpha + X, \beta + n - X)$. Sensible choices include Jeffreys prior $\text{Beta}(1/2, 1/2)$ and the uniform prior $\text{Beta}(1, 1)$.

The equal-tailed, Bayesian credible interval is obtained by taking the $\delta_{\text{est}}/2$ and $1 - \delta_{\text{est}}/2$ quantiles of $p \mid X$. Unlike those of Clopper-Pearson, these intervals are not inherently conservative. Indeed,

Clopper-Pearson intervals contain Jeffreys intervals [9]. This property is sometimes desirable, since it may mean that fewer samples are required to make a decision. In Bayesian optimization, however, one typically assumes that evaluating $f$ is far more expensive than simulating it. We therefore opted to use the more conservative choice.

Lastly, it should be said that bounds on estimation errors can also be obtained when $Z$ is not Bernoulli. For example, previous works [4, 33, 6] proposed to use an empirical Bernstein bound to generate confidence intervals for random variables $Z \in [a, b]$, defined here as

$$\left| \mathbb{E}[Z] - \overline{Z}_n \right| \leq \Delta_n = S_n \sqrt{\frac{2 \log(3/\delta_{\text{est}})}{n}} + \frac{3(b-a)\log(3/\delta_{\text{est}})}{n}, \tag{11}$$

where $\overline{Z}_n = \frac{1}{n} \sum_{i=1}^n z_i$ and $S_n^2 = \frac{1}{n} \sum_{i=1}^n \left( \overline{Z}_n - z_i \right)^2$ denote the empirical mean and variance. While conservative, the resulting intervals $\overline{Z}_n \pm \Delta_n$ decay much faster than, e.g., their Hoeffding-inequality-based counterparts when $S_n$ is much smaller than $b - a$.

Figure 4 illustrates how each of the methods discussed above perform in the context of Algorithm 2.

## A.2 How to choose where to evaluate the stopping rule

Per Algorithm 1, Algorithm 2 may be evaluated in parallel on a set of candidates $\mathbf{C} \subseteq \mathcal{X}$. This confers certain advantages, such as the ability to share draws of $f_t$ and, hence, $f_t^*$ between points $\boldsymbol{x} \in \mathbf{C}$. However, we must divide risk tolerance $\delta_{\text{est}}^t$ by cardinality $|\mathbf{C}|$ to retain the union bound. Hence, Algorithm 2 may be slow when $\mathbf{C}$ is large. We should therefore chose $\mathbf{C}$ with care.

If solutions must belong to the set of previously evaluate points, $\mathbf{X}_t$, then we suggest to define

$$\mathbf{C} = \{\boldsymbol{x} \in \mathbf{X}_t : \mathbb{P}(f_t(\boldsymbol{s}_t) - f_t(\boldsymbol{x}) \leq \epsilon) \geq 1 - \delta_{\text{mod}}\}, \tag{12}$$

since the excluded points can safely be ignored. Empirically, we found that this heuristic usually eliminates all but a few points.

If solutions may be chosen freely on $\mathcal{X}$, we instead recommend that $\mathbf{C}$ be constructed using one of the alternative estimators from Section 3.1 (all of which are differentiable). In particular, we recommend using gradient-based methods to maximize the average of

$$\mathbb{P}(r_t(\boldsymbol{x}) \leq \epsilon \mid f_t(\boldsymbol{x}) \leq f_t^*, f_t(\boldsymbol{x}_t^*) = f_t^*) = \Phi\left( \frac{f_t^* - \mu_{t+1}(\boldsymbol{x})}{k_{t+1}(\boldsymbol{x}, \boldsymbol{x})^{1/2}} \right)^{-1} \Phi\left( \frac{\mu_{t+1}(\boldsymbol{x}) - f_t^* - \epsilon}{k_{t+1}(\boldsymbol{x}, \boldsymbol{x})^{1/2}} \right), \tag{13}$$

over multiple draws of $f_t^*$ and $\boldsymbol{x}_t^* \in \arg\max_{\boldsymbol{x} \in \mathcal{X}} f_t(\boldsymbol{x})$, where $\mu_{t+1}$ and $k_{t+1}$ are the posterior mean and variance of $f_t$ given an additional observation $f(\boldsymbol{x}_t^*) = f_t^*$ and $\Phi$ denotes the standard normal cumulative distribution function. The resulting set of points can then be tested using Algorithm 2.

## A.3 How to schedule risk tolerances $\delta_{\text{est}}^t$

Where possible, we recommend using a (conservatively chosen) budget $T \in \mathbb{N}$ for BO. This mean we suggest using PRB together with a fixed budget. Doing so not only ensures that the algorithm stops in a reasonable amount of time, but allows one to use a constant schedule $\delta_{\text{est}}^t = \frac{\delta_{\text{est}}}{T - T_0}$, where $T_0 \in \mathbb{N}$ denotes the starting time. Note that this is how experiments from Section 5 were run.

If no such budget is available, then we recommend adopting a strategy similar to Algorithm 2 by only evaluating the stopping rule at certain steps $t$. For example, one may employ a geometric sequence of potential stopping times $(t_i)$, analogous to sample sizes $(n_j)$, and define

$$\delta_{\text{est}}^{t_i} = t_i^{-\alpha} \frac{(\alpha - 1)}{\alpha} \delta_{\text{est}} \qquad\qquad \alpha > 1, \tag{14}$$

like $d_t^j$ from Section 3.2. This practice ensures that $\delta_{\text{est}}^t$ does not decay too quickly.

# B  Technical proofs

The main results of this section are as follows.

   i. Proposition 4 shows that a family of acquisition functions produce dense sequences $(\boldsymbol{x}_t)$.

   ii. Lemma 6 proves that variances vanish as $\mathbf{X}_t$ becomes increasingly dense in $\mathcal{X}$.

   iii. Lemma 9 bounds the expected supremum of $f \sim \mathcal{GP}(0, k)$ in terms of its maximum variance.

   iv. Proposition 1 and Corollary 10 show that the PRB stopping criterion almost surely converges.

   v. Proposition 2 proves that BO with the PRB rule terminates and returns an $(\epsilon, \delta)$-optimal solution.

Many of the findings presented here and discussed previously borrow heavily from earlier works. Where appropriate, we attribute credit at the beginning of each proof.

**Definition 3.** *Kernel $k : \mathcal{X} \times \mathcal{X} \to \mathbb{R}$ has the no-empty-ball property [47] if, for any sequence $(\boldsymbol{x}_t)$, the posterior variance $\operatorname{Var}[f(\boldsymbol{x}) \mid f(\mathbf{X}_t)]$ at a point $\boldsymbol{x} \in \mathcal{X}$ goes to zero as $t \to \infty$ if and only if $\boldsymbol{x}$ is an adherent point of $\{\boldsymbol{x}_t : t \geq 0\}$.*

**Proposition 4.** *Let $f \sim \mathcal{GP}(0, k)$ be a prior over functions on a compact space $\mathcal{X} \in \mathbb{R}^D$ and $V_t : \mathcal{X} \to \mathbb{R}$ be a continuous acquisition function for $f_t$. If $k$ is a continuous kernel that admits the no-empty-ball property and*

$$k_t(\boldsymbol{x}, \boldsymbol{x}) > k_t(\boldsymbol{x}', \boldsymbol{x}') = 0 \implies V_t(\boldsymbol{x}) > V_t(\boldsymbol{x}') \qquad \forall t \in \mathbb{N} \ \text{and} \ \forall \boldsymbol{x}, \boldsymbol{x}' \in \mathcal{X}, \quad (15)$$

*then the sequence $(\boldsymbol{x}_t)$ of points $\boldsymbol{x}_t \in \arg\max_{\boldsymbol{x} \in \mathcal{X}} V_t(\boldsymbol{x})$ is dense in $\mathcal{X}$.*

*Proof.* Follows immediately from [47]. Without loss of generality, suppose $V_t(\boldsymbol{x})$ is non-negative and equals zero if and only if $k_t(\boldsymbol{x}, \boldsymbol{x}) = 0$. Let $(\boldsymbol{x}_{a_t})$ and $(\boldsymbol{x}_{b_t})$ be subsequence of $(\boldsymbol{x}_t)$ that converge to an accumulation point $\boldsymbol{z} \in \mathcal{X}$ and write $\alpha_t = \max\{a_i : a_i \leq t\}$ and $\beta_t = \max\{b_i : b_i \leq t\}$. Then,

$$\begin{aligned}
\operatorname{Var}[f(\boldsymbol{x}_{\alpha_t}) \mid f(\mathbf{X}_t)] &\leq \operatorname{Var}[f(\boldsymbol{x}_{\alpha_t}) \mid f(\boldsymbol{x}_{\beta_t})] \\
&\leq k(\boldsymbol{x}_{\alpha_t}, \boldsymbol{x}_{\alpha_t}) + k(\boldsymbol{x}_{\beta_t}, \boldsymbol{x}_{\beta_t}) - 2k(\boldsymbol{x}_{\alpha_t}, \boldsymbol{x}_{\beta_t}).
\end{aligned} \quad (16)$$

Since $(\boldsymbol{x}_{a_t})$ and $(\boldsymbol{x}_{b_t})$ both converge to $\boldsymbol{z}$ and $k$ is assumed continuous, (16) goes to zero as $t \to \infty$. Consequently, $V_{a_t}(\boldsymbol{x}_{a_t})$ and, therefore, $V_t(\boldsymbol{x}_t)$ must also vanish as $t \to \infty$ [47, Proposition 12]. By definition of $\boldsymbol{x}_t$, it follows that $\lim_{t \to \infty} \max_{\boldsymbol{x} \in \mathcal{X}} k_t(\boldsymbol{x}, \boldsymbol{x}) = 0$. The no-empty-ball property now gives the result. $\qquad \square$

**Proposition 5.** *Let $\mathcal{X} \subseteq \mathbb{R}^D$ be convex and suppose that $\mathbf{X} \subseteq \mathcal{X}$ generates an $\varepsilon$-cover of $\mathcal{X}$. For every $\boldsymbol{x} \in \mathcal{X}$ and $\rho \geq \varepsilon$, the intersection of the set $\mathbf{X}$ and the ball $B(\boldsymbol{x}, \rho) = \{\boldsymbol{x}' \in \mathcal{X} : \|\boldsymbol{x} - \boldsymbol{x}'\|_\infty \leq \rho\}$ generates a $2\varepsilon$-cover of $B(\boldsymbol{x}, \rho)$.*

*Proof.* Consider the ball $B(\boldsymbol{x}, r)$ with radius $r = \rho - \varepsilon$. Since $\mathcal{X}$ is convex, for every point $\boldsymbol{a} \in B(\boldsymbol{x}, \rho)$ there exists a $\boldsymbol{b} \in B(\boldsymbol{x}, r)$ such that $\|\boldsymbol{a} - \boldsymbol{b}\|_\infty \leq \varepsilon$. Moreover, because $\mathbf{X}$ generates an $\varepsilon$-cover of $\mathcal{X}$, for every point $\boldsymbol{b} \in B(\boldsymbol{x}, r)$ there exists a $\boldsymbol{c} \in \mathbf{X}$ so that $\|\boldsymbol{b} - \boldsymbol{c}\|_\infty \leq \varepsilon$, which implies that $\boldsymbol{c} \in B(\boldsymbol{x}, \rho)$. It follows by the triangle inequality that for every point $\boldsymbol{a} \in B(\boldsymbol{x}, \rho)$ there exists a pair of points $\boldsymbol{b}, \boldsymbol{c} \in B(\boldsymbol{x}, r) \times [B(\boldsymbol{x}, \rho) \cap \mathbf{X}]$ such that

$$\|\boldsymbol{a} - \boldsymbol{c}\|_\infty \leq \|\boldsymbol{a} - \boldsymbol{b}\|_\infty + \|\boldsymbol{b} - \boldsymbol{c}\|_\infty \leq \varepsilon + \varepsilon = 2\varepsilon, \quad (17)$$

which completes the proof. $\qquad \square$

**Lemma 6.** *Under assumptions A1 and A2, if $y(\cdot) \sim \mathcal{N}\big(f(\cdot), \gamma^2\big)$ is observed on a set of points $\mathbf{X} \subseteq \mathcal{X}$ that generates an $\varepsilon$-cover of $\mathcal{X}$, $0 \leq \varepsilon \leq \min\{1, {}^{k(\boldsymbol{x},\boldsymbol{x})}/_{L_k}\}$, then*

$$\operatorname{Var}[f(\boldsymbol{x}) \mid y(\mathbf{X})] \leq \kappa_\varepsilon(\boldsymbol{x}), \quad (18)$$

*where*

$$\kappa_\varepsilon(\boldsymbol{x}) = \frac{\big[4L_k \rho(\varepsilon) k(\boldsymbol{x}, \boldsymbol{x}) - L_k^2 \rho(\varepsilon)^2\big] \eta(\varepsilon) + \gamma^2 k(\boldsymbol{x}, \boldsymbol{x})}{[k(\boldsymbol{x}, \boldsymbol{x}) + 2L_k \rho(\varepsilon)] \eta(\varepsilon) + \gamma^2}$$

*is given in terms of $\eta(\varepsilon) = \max\{1, {}^{\rho(\varepsilon)}/_{4\varepsilon}\}^D$ and $\rho(\varepsilon) = \varepsilon^\varepsilon$ for any $0 < \varepsilon < 1$.*

*Proof.* This result extends Lederer et al. [26, Theorem 3.1], who showed that, for all $0 \leq \rho \leq {k(\boldsymbol{x},\boldsymbol{x})}/{L_k}$,

$$\text{Var}[f(\boldsymbol{x}) \mid y(\mathbf{B}_\rho(\boldsymbol{x}))] \leq \frac{(4L_k\rho k(\boldsymbol{x},\boldsymbol{x}) - L_k^2\rho^2)|\mathbf{B}_\rho(\boldsymbol{x})| + \gamma^2 k(\boldsymbol{x},\boldsymbol{x})}{(k(\boldsymbol{x},\boldsymbol{x}) + 2L_k\rho)|\mathbf{B}_\rho(\boldsymbol{x})| + \gamma^2}, \tag{19}$$

where $|\mathbf{B}_\rho(\boldsymbol{x})|$ is the cardinality of the set $\mathbf{B}_\rho(\boldsymbol{x}) = B(\boldsymbol{x},\rho) \cap \mathbf{X}$. We would like to convert this upper bound into a function of $0 \leq \varepsilon \leq 1$. To this end, begin by noticing that the bound (19) increases monotonically on $0 \leq \rho \leq {k(\boldsymbol{x},\boldsymbol{x})}/{L_k}$ and decreases monotonically on $n = |\mathbf{B}_\rho(\boldsymbol{x})| \in \mathbb{N}_0$. Substituting $\rho(\varepsilon)$ for $\rho$ and $\eta(\varepsilon)$ for $n$ therefore yields a valid bound so long as $\rho \leq \rho(\varepsilon) \leq {k(\boldsymbol{x},\boldsymbol{x})}/{L_k}$ and $0 \leq \eta(\varepsilon) \leq n$. For clarity, note that $\rho(\varepsilon)$ defines the radius of a ball around $\boldsymbol{x}$ and $\eta(\varepsilon)$ denotes the minimum possible number of elements from $\mathbf{X}$ that lie within this ball.

Starting with the latter, lower bounds on the cardinality of $\mathbf{B}_\rho(\boldsymbol{x})$ may be obtained from the fact that $\mathbf{X}$ is assumed to generate an $\varepsilon$-cover of $\mathcal{X}$. By Proposition 5, it follows that $\mathbf{B}_\rho(\boldsymbol{x})$ generates a $2\varepsilon$-cover of $B(\boldsymbol{x},\rho)$. Accordingly, $|\mathbf{B}_\rho(\boldsymbol{x})|$ must be greater-equal to the minimum number of points required to construct such a cover. Under the $\|\cdot\|_\infty$ norm, the $\varepsilon$-covering number of a ball

$$B(\boldsymbol{x},\rho) = \prod_{d=1}^{D}[\max(x_d - \rho, 0), \min(x_d + \rho, 1)] \tag{20}$$

is given by

$$M\big(B(\boldsymbol{x},\rho), \|\cdot\|_\infty, \varepsilon\big) = \prod_{d=1}^{D}\left\lceil \frac{\min(x_d + \rho, 1) - \max(0, x_d - \rho)}{2\varepsilon} \right\rceil. \tag{21}$$

This number is minimized when $B(\cdot,\rho)$ is placed in a corner, such as $B(\mathbf{0},\rho) = [0,\rho]^D$. Choosing

$$\eta(\varepsilon) = \max\left\{1, \left(\frac{\rho(\varepsilon)}{4\varepsilon}\right)^D\right\} \leq \left\lceil \frac{\rho(\varepsilon)}{4\varepsilon} \right\rceil^D \tag{22}$$

therefore ensures that $\eta(\varepsilon)$ lower bounds the cardinality of every $\mathbf{B}_\rho(\cdot)$. Note that there are two factors of two at play here: one accounts for the fact that $\mathbf{B}_\rho(\cdot)$ is only guaranteed to provide a $2\varepsilon$-cover of $B(\cdot,\rho)$, and the other accounts for the fact that the corner balls are up to $2^D$ times smaller than other balls with the same radius.

Turning our attention to the choice of function $\rho(\varepsilon)$, some desiderata come into focus. First, we require $\rho(\varepsilon) \geq \varepsilon$ so that every $\mathbf{B}_\rho(\cdot)$ is nonempty. Second, we desire $\lim_{\varepsilon \to 0^+} \rho(\varepsilon) = 0$ because the resulting posterior variance bound will increase monotonically in $\rho(\varepsilon)$. Lastly, we want the ratio of $\rho(\varepsilon)$ to $\varepsilon$ to diverge to infinity as $\varepsilon$ approaches zero from above so that $\lim_{\varepsilon \to 0^+} \eta(\varepsilon) = \infty$. Based on these criteria, a convenient choice when $\mathcal{X} = [0,1]^D$ is

$$\rho(\varepsilon) = \varepsilon^\alpha \qquad\qquad 0 < \alpha < 1. \tag{23}$$

In summary, the claim follows by expressing $\rho$ as a function of $\varepsilon$ and using it to lower bound $|\mathbf{B}_\rho(\cdot)|$ with $\eta(\varepsilon)$:

$$\text{Var}[f(\boldsymbol{x}) \mid y(\mathbf{X})] \leq \frac{(4L_k\rho(\varepsilon)k(\boldsymbol{x},\boldsymbol{x}) - L_k^2\rho(\varepsilon)^2)\eta(\varepsilon) + \gamma^2 k(\boldsymbol{x},\boldsymbol{x})}{(k(\boldsymbol{x},\boldsymbol{x}) + 2L_k\rho(\varepsilon))\eta(\varepsilon) + \gamma^2}. \tag{24}$$

$\square$

**Proposition 7.** *For any choice of constants $a > 0$, $b \geq 0$, $c \geq 0$,*

$$\int_0^c \sqrt{\log(1 + b\varepsilon^{-1/a})}d\varepsilon \leq c\sqrt{a^{-1} + \log\big(1 + bc^{-1/a}\big)}. \tag{25}$$

*Proof.* This proof ammends Grünewälder et al. [18, Appendix A]. Let $\xi = (1 + \sqrt[a]{c}b^{-1})^a$ so that

$$\int_0^c \sqrt{\log\big(1 + b\varepsilon^{-1/a}\big)}d\varepsilon \leq \int_0^c \sqrt{\log\big(\xi^{1/a}b\varepsilon^{-1/a}\big)}d\varepsilon. \tag{26}$$

Next, define auxiliary functions

$$f(u) = \sqrt{\log(u^{-1/a})} \qquad\qquad g(\varepsilon) = \frac{\varepsilon}{\xi b^a} \tag{27}$$

such that $f(g(\varepsilon)) = \sqrt{\log(\xi^{1/a} b \varepsilon^{-1/a})}$ and use them to integrate by substitution as

$$\int_0^c \sqrt{\log(\xi^{1/a} b \varepsilon^{-1/a})}\, d\varepsilon = \xi b^a \int_0^{g(c)} \sqrt{\log(u^{-1/a})}\, du = \frac{\xi b^a}{\sqrt{a}} \int_0^{g(c)} \sqrt{-\log(u)}\, du. \tag{28}$$

The Cauchy-Schwarz inequality now gives

$$\int_0^{g(c)} \sqrt{-\log(u)}\, du \leq \left( \int_0^{g(c)} du \right)^{1/2} \left( -\int_0^{g(c)} \log(u)\, du \right)^{1/2} = \frac{c}{\xi b^a} \sqrt{1 - \log\left( \frac{c}{\xi b^a} \right)}. \tag{29}$$

Hence, the claim follows

$$\int_0^c \sqrt{\log(1 + b\varepsilon^{-1/a})}\, d\varepsilon \leq c\sqrt{\frac{1 + \log(\xi b^a c^{-1})}{a}} = c\sqrt{a^{-1} + \log(1 + bc^{-1/a})}. \tag{30}$$

$\square$

**Remark 8.** *For comparison with Grünewälder et al. [18], if $b^a = 2c$ then Proposition 7 gives*

$$\xi = \left( 1 + 2^{-1/a} \right)^a \leq 2^a \implies c\sqrt{\frac{1 + \log(\xi b^a c^{-1})}{a}} = c\sqrt{\frac{1 + \log(2\xi)}{a}} \leq c\sqrt{\frac{\log(e2^{a+1})}{a}}, \tag{31}$$

*which matches their reported result.*

**Lemma 9.** *Let $f \sim \mathcal{GP}(0, k)$ be a Gaussian process with an $L_k$-Lipschitz continuous covariance function $k : \mathcal{X}^2 \to \mathbb{R}$ on $\mathcal{X} = [0, r]^D$ having maximum variance $\sigma^2 = \max_{\boldsymbol{x} \in \mathcal{X}} k(\boldsymbol{x}, \boldsymbol{x})$. Then,*

$$\mathbb{E}\left[ \sup_{\boldsymbol{x} \in \mathcal{X}} f(\boldsymbol{x}) \right] \leq 12\sigma\sqrt{2D + D\log(1 + 4L_k r\sigma^{-2})}. \tag{32}$$

*Proof.* This proof paraphrases parts of Grünewälder et al. [18, Section 4.3].

Massart [30, Theorem 3.18] proved that the expected supremum of $f$ is upper bounded by

$$\mathbb{E}\left[ \sup_{\boldsymbol{x} \in \mathcal{X}} f(\boldsymbol{x}) \right] \leq 12 \int_0^\sigma \sqrt{\log N(\mathcal{X}, d_k, \varepsilon)}\, d\varepsilon, \tag{33}$$

where $N(\mathcal{X}, d_k, \varepsilon)$ is defined as the $\varepsilon$-packing number—i.e. the largest number of points that can be "packed" inside of $\mathcal{X}$ without any two points being within $\varepsilon$ of one another—under the canonical pseudo-metric[3]

$$d_k(\boldsymbol{x}, \boldsymbol{x}') = \mathbb{E}\big[(f(\boldsymbol{x}) - f(\boldsymbol{x}'))^2\big]^{1/2} = \sqrt{k(\boldsymbol{x}, \boldsymbol{x}) - 2k(\boldsymbol{x}, \boldsymbol{x}') + k(\boldsymbol{x}', \boldsymbol{x}')}. \tag{34}$$

We may use (33) by upper bounding the right-hand side with a known quantity. We will bound the $\epsilon$-packing number $N(\mathcal{X}, d_k, \varepsilon)$, translate this bound from the $d_k$ pseudo-metric to the infinity norm, and then integrate the result.

The first step follows immediately from the fact that the $\varepsilon$-packing number is smaller than the $\frac{\varepsilon}{2}$-covering number—defined as the minimum number of balls $B(\cdot, \frac{\varepsilon}{2})$ required to cover $\mathcal{X}$. The second is accomplished by using Lipschitz continuity of $k$ to show that the squared pseudo-metric $d_k(\cdot, \cdot)^2$ is $2L_k$-Lipschitz: for all $\boldsymbol{x}, \boldsymbol{x}' \in \mathcal{X}$,

$$d_k(\boldsymbol{x}, \boldsymbol{x}')^2 = \big[k(\boldsymbol{x}, \boldsymbol{x}) - k(\boldsymbol{x}, \boldsymbol{x}')\big] + \big[k(\boldsymbol{x}', \boldsymbol{x}') - k(\boldsymbol{x}', \boldsymbol{x})\big] \leq 2L_k \|\boldsymbol{x} - \boldsymbol{x}'\|_\infty. \tag{35}$$

It follows that, for any set $\mathbf{X} \subseteq \mathcal{X}$,

$$\max_{\boldsymbol{x} \in \mathcal{X}} \min_{\boldsymbol{x}' \in \mathbf{X}} \|\boldsymbol{x} - \boldsymbol{x}'\|_\infty \leq C \implies \max_{\boldsymbol{x} \in \mathcal{X}} \min_{\boldsymbol{x}' \in \mathbf{X}} d_k(\boldsymbol{x}, \boldsymbol{x}') \leq \sqrt{2L_k C}. \tag{36}$$

---

[3]While $d_k$ has most of the properties of a proper metric, $d_k(\boldsymbol{x}, \boldsymbol{x}') = 0$ need not always imply $\boldsymbol{x} = \boldsymbol{x}'$ [2].

An $\varepsilon^2/8L_k$-cover under the infinity norm therefore guarantees an $\varepsilon/2$-cover under $d_k$. The former may be constructed from a grid of uniformly spaced points with elements at intervals of $\varepsilon^2/4L_k$. This grid will consist of $\lceil 4L_k r\varepsilon^{-2}\rceil^D$ points assuming $\mathcal{X} = [0, r]^D$, meaning that

$$N(\mathcal{X}, d_k, \varepsilon) < \left(1 + 4L_k r\varepsilon^{-2}\right)^D. \tag{37}$$

To complete the proof, use Proposition 7 with $a = \frac{1}{2}$, $b = 4L_k r$, and $c = \sigma$ to show that

$$\mathbb{E}\left[\sup_{\boldsymbol{x}\in\mathcal{X}} f(\boldsymbol{x})\right] \leq 12\sqrt{D}\int_0^\sigma \sqrt{\log(1 + 4L_k r\varepsilon^{-2})}d\varepsilon \leq 12\sigma\sqrt{2D + D\log(1 + 4L_k r\sigma^{-2})}. \tag{38}$$

$\square$

**Proposition 1.** *Under assumptions A1–A3 and for all regret bounds $\epsilon > 0$ and risk tolerances $\delta > 0$, there almost surely exists $T \in \mathbb{N}_0$ so that, at each time $t \geq T$, every $\boldsymbol{s}_t \in \arg\max_{\boldsymbol{x}\in\mathcal{X}} \mu_t(\boldsymbol{x})$ satisfies*

$$\Psi_t(\boldsymbol{x}; \epsilon) = \mathbb{P}(r_t(\boldsymbol{s}_t) \leq \epsilon) \geq 1 - \delta. \tag{8}$$

*Proof.* Consider the centered process

$$g_t(\cdot) = \left[f_t(\cdot) - \mu_t(\cdot)\right] - \left[f_t(\boldsymbol{s}_t) - \mu_t(\boldsymbol{s}_t)\right], \tag{39}$$

with covariance

$$c_t(\boldsymbol{x}, \boldsymbol{x}') = k_t(\boldsymbol{x}, \boldsymbol{x}') - k_t(\boldsymbol{x}, \boldsymbol{s}_t) - k_t(\boldsymbol{s}_t, \boldsymbol{x}') + k_t(\boldsymbol{s}_t, \boldsymbol{s}_t). \tag{40}$$

The term $\mu_t(\boldsymbol{s}_t) - \mu_t(\cdot)$ is nonnegative by construction such that

$$g_t^* = \sup_{\boldsymbol{x}\in\mathcal{X}} g_t(\boldsymbol{x}) \geq f_t^* - f_t(\boldsymbol{s}_t) \qquad\qquad f_t^* = \sup_{\boldsymbol{x}\in\mathcal{X}} f_t(\boldsymbol{x}) \tag{41}$$

and, therefore,

$$\mathbb{P}(g_t^* \geq \epsilon) \geq \mathbb{P}(f_t^* - f_t(\boldsymbol{s}_t) \geq \epsilon). \tag{42}$$

We would now like to use the Borell-TIS inequality [8, 46] to show that: if $\epsilon > \mathbb{E}(g_t^*)$, then

$$\mathbb{P}(g_t^* \geq \epsilon) \leq \exp\left(-\frac{1}{2}\left[\frac{\epsilon - \mathbb{E}(g_t^*)}{2\sigma_t}\right]^2\right), \tag{43}$$

where $\sigma_t = \max_{\boldsymbol{x}\in\mathcal{X}} \sqrt{k_t(\boldsymbol{x}, \boldsymbol{x})}$ and $2\sigma_t$ appears in the denominator (rather than $\sigma_t$) because $\max_{\boldsymbol{x}\in\mathcal{X}} c_t(\boldsymbol{x}, \boldsymbol{x}) \leq 4\sigma_t^2$. Since (43) is an increasing, continuous function of both $\sigma_t \geq 0$ and $0 \leq \mathbb{E}(g_t^*) < \epsilon$, the claim will hold if these quantities vanish as the (global) fill distance $h_t = \max_{\boldsymbol{x}\in\mathcal{X}} \min_{1\leq i\leq t} \|\boldsymbol{x} - \boldsymbol{x}_i\|_\infty$ goes to zero.

The former result is an immediate consequence of Lemma 6. Regarding the latter, $f_t(\boldsymbol{s}_t) - \mu_t(\boldsymbol{s}_t)$ is a centered random variable. It follows by linearity of expectation that

$$\mathbb{E}(g_t^*) = \mathbb{E}\left[\sup_{\boldsymbol{x}\in\mathcal{X}} f_t(\boldsymbol{x}) - \mu_t(\boldsymbol{x})\right]. \tag{44}$$

Next, denote the canonical pseudo-metric at time $t$ by

$$d_{k_t}(\boldsymbol{x}, \boldsymbol{x}') = \mathbb{E}\left[(f_t(\boldsymbol{x}) - f_t(\boldsymbol{x}'))^2\right]^{1/2} = \sqrt{k_t(\boldsymbol{x}, \boldsymbol{x}) - 2k_t(\boldsymbol{x}, \boldsymbol{x}') + k_t(\boldsymbol{x}', \boldsymbol{x}')}. \tag{45}$$

This pseudo-metric is non-increasing in $t$. To see this, let $\beta = k(\boldsymbol{x}_{t+1}, \boldsymbol{x}) - k(\boldsymbol{x}_{t+1}, \boldsymbol{x}')$ and write

$$d_{k_{t+1}}(\boldsymbol{x}, \boldsymbol{x}')^2 = k_{t+1}(\boldsymbol{x}, \boldsymbol{x}) - 2k_{t+1}(\boldsymbol{x}, \boldsymbol{x}') + k_{t+1}(\boldsymbol{x}', \boldsymbol{x}')$$

$$= \underbrace{k_t(\boldsymbol{x}, \boldsymbol{x}) - 2k_t(\boldsymbol{x}, \boldsymbol{x}') + k_t(\boldsymbol{x}', \boldsymbol{x}')}_{d_{k_t}(\boldsymbol{x}, \boldsymbol{x}')^2} - \underbrace{\beta^2\left[k_t(\boldsymbol{x}_{t+1}, \boldsymbol{x}_{t+1}) + \gamma^2\right]^{-1}}_{\geq 0}. \tag{46}$$

As $t$ increases, points therefore become closer together under the $d_{k_t}$ pseudo-metric. For this reason, the posterior $\varepsilon$-packing number $N(\mathcal{X}, d_{k_t}, \varepsilon)$ is less-equal to the prior $\varepsilon$-packing number $N(\mathcal{X}, d_k, \varepsilon)$. By Lemma 9, we now have

$$\mathbb{E}(g_t^*) \leq \int_0^{\sigma_t} \sqrt{\log N(\mathcal{X}, d_{k_t}, \varepsilon)}d\varepsilon$$

$$\leq \int_0^{\sigma_t} \sqrt{\log N(\mathcal{X}, d_k, \varepsilon)}d\varepsilon \tag{47}$$

$$\leq 12\sigma_t\sqrt{2D + D\log\left(1 + 4L_k\sigma_t^{-2}\right)}.$$

From here, note that (47) is an increasing, continuous function of $\sigma_t$ that vanishes as $\sigma_t \to \infty$. By Lemma 6, the same is true of $\sigma_t$ as a function of $h_t$. As a result, (43) becomes arbitrarily small as $h_t \to 0$ and there exists a constant $h_* > 0$ such that this upper bound is less-equal to $\delta$ whenever $h_t \leq h_*$. Finally, since $(\boldsymbol{x}_t)$ is almost surely dense in $\mathcal{X}$, there almost surely exists a time $T \in \mathbb{N}_0$ such that

$$t \geq T \implies h_t \leq h_* \implies \mathbb{P}(g_t^* > \epsilon) \leq \delta \implies \mathbb{P}(f_t^* - f_t(\boldsymbol{s}_t) > \epsilon) \leq \delta. \tag{48}$$

$\square$

**Corollary 10.** *Suppose assumptions A1–A3 hold and that there exists a constant $\epsilon' > 0$ so that*

$$\lim_{t \to \infty} \left[ \max_{\boldsymbol{x} \in \mathcal{X}} \mu_t(\boldsymbol{x}) - \mu_t(\boldsymbol{s}_t) \right] \leq \epsilon' \tag{49}$$

*with probability one, where $\boldsymbol{s}_t \in \arg\max_{\boldsymbol{x} \in \mathbf{X}_t} \mu_t(\boldsymbol{x})$. Then, for every $\epsilon > \epsilon'$ and $\delta \in (0, 1]$, there almost surely exists a time $T \in \mathbb{N}$ such that, for all $t \geq T$,*

$$\mathbb{P}\left[ \sup_{\boldsymbol{x} \in \mathcal{X}} f_t(\boldsymbol{x}) - f_t(\boldsymbol{s}_t) \leq \epsilon \right] \geq 1 - \delta. \tag{50}$$

*Proof.* Per (49), there almost surely exists an $S \in \mathbb{N}$ such that $t \geq S \implies \max_{\boldsymbol{x} \in \mathcal{X}} \mu_t(\boldsymbol{x}) - \mu_t(\boldsymbol{s}_t) \leq \epsilon'$. Proposition 1 therefore implies there almost surely exists a $T \geq S$ so that

$$t \geq T \implies \mathbb{P}\left[ f_t^* - f_t(\boldsymbol{s}_t) \geq \epsilon - \epsilon' \right] \leq \delta, \tag{51}$$

which completes the proof. $\square$

The assumption that the posterior mean approaches its maximum on $(\boldsymbol{x}_t)$ protects against adversarial cases where—no matter how densely we observe $f$—there is always an $\boldsymbol{x} \in \mathcal{X} \setminus \mathbf{X}_t$ so that $\mu_t(\boldsymbol{x})$ exceeds $\mu_t(\boldsymbol{s}_t)$ by at least $\epsilon$. Note that (49) becomes a necessary condition when $\delta < \frac{1}{2}$. Nevertheless, it is unclear how to ensure this condition without making stronger assumptions for $f$ and $(\boldsymbol{x}_t)$. One can use A2 and the Cauchy-Schwarz inequality to show that the posterior mean is Lipschitz continuous [27]; but, its Lipschitz constant may continue to grow as $t \to \infty$, so (49) may not hold.

## C   Experiment details

Experiments were run using a combination of GPFlow [31] and Trieste [36]. Runtimes reported in Figure 3 were measured on an Apple M1 Pro Chip using an off-the-shelf build of TensorFlow [1].

### C.1   Model specification

We employed Gaussian process priors $f \sim \mathcal{GP}(\mu, k)$ with constant mean functions $\mu(\cdot) = c$ and Matérn-$5/2$ covariance functions equipped with ARD lengthscales.

**True**   When optimizing functions drawn from GP priors, we set the prior mean to zero and used unit variance kernels with lengthscales $\ell_i = \frac{1}{4}\sqrt{D}$. Noise variances are reported alongside results.

**MAP**   When optimizing black-box functions, we employed broad and uninformative hyperpriors. Let $[\mathcal{X}]_i = [a_i, b_i]$ be the range of the $i$-th design variable, $q_t : [0, 1] \to \mathbb{R}$ be the empirical quantile function of $y$ at time $t$, and $\nu_t = \overline{\mathrm{Var}}[\boldsymbol{y}_{t-1}]$ be the empirical variance of observations $\boldsymbol{y}_{t-1} = \{y(\boldsymbol{x}_1), \ldots, y(\boldsymbol{x}_{t-1})\}$. Our hyperpriors are then as follows:

| Name | Distribution | Parameters | |
|---|---|---|---|
| Constant Mean | $\mathrm{Uniform}(a, b)$ | $a = q_t(0.05)$ | $b = q_t(0.95)$ |
| Log Kernel Variance | $\mathrm{Uniform}(a, b)$ | $a = \log(10^{-1}\nu_t)$ | $b = \log(10\nu_t)$ |
| Log Noise Variance | $\mathrm{Uniform}(a, b)$ | $a = \log(10^{-9}\nu_t)$ | $b = \log(10\nu_t)$ |
| $i$-th Lengthscale | $\mathrm{LogNormal}(\mu, \sigma)$ | $\mu = \frac{1}{2}(b_i - a_i)$ | $\sigma = 1$ |

Note that we directly parameterize certain hyperparamters in log-space and that, e.g., $\log(\theta) \sim \mathrm{Uniform}(a, b)$ is not the same as $\theta \sim \mathrm{LogUniform}(e^a, e^b)$.

## C.2 Link function

When modeling classification rates for MNIST and Adult, we used a logit (i.e. inverse sigmoid) link function,

$$g(y) = \log\left(\frac{y}{1-y}\right) \qquad\qquad g^{-1}(x) = \frac{1}{1 + e^{-x}}, \tag{52}$$

in order so that $g^{-1} \circ f : \mathcal{X} \to [0, 1]$. When evaluating stopping rules, we handled this link functions by pulling draws of, e.g., $f_t(\boldsymbol{x})$ backward through $g$ and using te resulting values to estimate expectations and probabilities. This approach was used for all but $\Delta$CB [29], where we instead computed $g^{-1} \circ \mathrm{UCB}_t$ and $g^{-1} \circ \mathrm{LCB}_t$.

## C.3 Acquisition function

In our experiments, we defined the set of feasible solutions at time $t \in \mathbb{N}$ as the set of previously evaluated points $\mathbf{X}_t$. Under these circumstances, one can show that the optimal one-step policy is given by an "in-sample" version of the Knowledge Gradient strategy [34, 15]. Let

$$\mu_{t+1}(\cdot; \boldsymbol{x}, z) = \mu_t(\cdot) + \frac{k_t(\cdot, \boldsymbol{x})z}{\sqrt{k_t(\boldsymbol{x}, \boldsymbol{x}) + \gamma^2}} \qquad \sigma_{t+1}(\cdot; \boldsymbol{x}) = \sigma_t(\cdot) - \frac{k_t(\cdot, \boldsymbol{x})^2}{k_t(\boldsymbol{x}, \boldsymbol{x}) + \gamma^2} \tag{53}$$

be the posterior mean and variance of $f$ at time $t+1$ if we observe $y_{t+1} = \mu_t(\boldsymbol{x}) + z\sqrt{k_t(\boldsymbol{x}, \boldsymbol{x}) + \gamma^2}$, where $z \sim \mathcal{N}(0, 1)$. Further, at times $t$ and $t+1$ define

$$\nu_t(\boldsymbol{x}) = \mathbb{E}[(g^{-1} \circ f_t)(\boldsymbol{x})] = \mathbb{E}[g^{-1}(\xi)] \qquad\qquad \xi \sim \mathcal{N}(\mu_t(\boldsymbol{x}), \sigma_t^2(\boldsymbol{x})) \tag{54}$$

as the corresponding expected value when accounting for a link function. If no link function is given, then $\nu_t(\cdot) = \mu_t(\cdot)$. Then, the aforementioned acquisition function is given by

$$\mathrm{ISKG}_t(\boldsymbol{x}) = \mathbb{E}_z\left[\max \nu_{t+1}(\mathbf{X}_t \cup \{\boldsymbol{x}\}; \boldsymbol{x}, z)\right] - \max \nu_t(\mathbf{X}_t) \qquad z \sim \mathcal{N}(0, 1). \tag{55}$$

ISKG is identical to the Expected Improvement function when $\gamma^2 = 0$ [39], but avoids pathologies (such as re-evaluating previously observed points) when $\gamma^2 > 0$.

In practice, we estimated (55) with Gauss-Hermite quadrature and maximized it using multi-start gradient ascent [50, 5]. Likewise, we either evaluated (54) analytically or via quadrature. Starting positions we obtained by running CMA-ES [19] several times to partial convergence. The best point from each run was then combined with a large number of random points and the top 16 points were fine-tuned using L-BFGS-B [10].

## C.4 Convolutional neural networks

| Name | Low | High |
|------|-----|------|
| Num. filters | 1 | 64 |
| Num. epochs | 1 | 25 |
| Log learning rate | $\log(10^{-5})$ | 0 |
| Dropout rate | 0 | 1 |

When training convolutional neural networks (CNNs) on MNIST [14], we used a simple architecture consisting of two convolutional layers with $3 \times 3$ filters and ReLU activation functions [3] followed by max pooling layers with a pool-size of 2. The output of the final pooling layer was flattened and subjected to dropout before being passed to a dense classification layer consisting of ten neurons. Each model was trained using Adam [24], with batches of size 64. The search space for this problem is depicted on the right. Integer valued parameters were handled by rounding to the nearest value. To obtain a reliable estimate of the minimum achievable misclassification rate, the same random seed was used for each training run.

## C.5 XGBoost classifiers

| Name | Low | High |
|------|-----|------|
| Max. tree depth | 1 | 10 |
| Log num. estimators | 0 | $\log(10^3)$ |
| Log learning rate | $\log(10^{-3})$ | 0 |

We used an off-the-shelf implementation of XGBoost [12] for the the adult income classification problem [7]. The search space was three-dimensional and is shown on the right. Integer valued parameters were handled by rounding to the nearest value. To obtain a reliable estimate of the minimum achievable misclassification rate, the same random seed was used to when generating train-test splits and for each training run.

# D  Extended results

## D.1  Results without adjusted cutoff values

| Problem | $D$ | $T$ | **Oracle**[†] | **Budget**[†] | **Acq** | **ΔCB** | **ΔES** | **PRB** |
|---|---|---|---|---|---|---|---|---|
| **GP**[†] $10^{-6}$ | 2 | 64 | 10 (100) | 17 (96) | 28 (100) | 12 (89) | 14 (94) | **17 (97)** |
| **GP**[†] $10^{-2}$ | 2 | 128 | 11 (100) | 22 (96) | 78 (100) | 82 (100) | 18 (91) | **23 (99)** |
| **GP**[†] $10^{-6}$ | 4 | 128 | 27 (100) | 64 (95) | 90 (100) | 23 (66) | 28 (74) | **64 (99)** |
| **GP**[†] $10^{-2}$ | 4 | 256 | 30 (100) | 94 (95) | 106 (98) | 256 (100) | 36 (65) | **86 (96)** |
| **GP**[†] $10^{-6}$ | 6 | 256 | 40 (99) | 124 (95) | 142 (98) | 31 (50) | 46 (65) | **134 (98)** |
| **GP**[†] $10^{-2}$ | 6 | 512 | 65 (100) | 227 (96) | **181 (96)** | 512 (100) | 45 (34) | 235 (100) |
| **GP** $10^{-6}$ | 4 | 128 | 35 (100) | 79 (95) | **92 (100)** | 18 (30) | 22 (41) | 61 (88) |
| **GP** $10^{-2}$ | 4 | 256 | 51 (100) | 157 (95) | **128 (97)** | 224 (80) | 27 (22) | 100 (92) |
| **Branin** | 2 | 128 | 19 (100) | 25 (95) | 64 (100) | **31 (99)** | 32 (100) | 33 (99) |
| **Hartmann** | 3 | 64 | 14 (100) | 22 (96) | 26 (100) | 15 (83) | 17 (84) | **19 (100)** |
| **Hartmann** | 6 | 64 | 36 (67) | 256 (67) | 40 (67) | 26 (46) | 30 (56) | 40 (64) |
| **Rosenbrock** | 4 | 96 | 34 (100) | 46 (95) | 95 (100) | **68 (99)** | 71 (100) | 84 (100) |
| **CNN** | 4 | 256 | 5 (100) | 11 (96) | 64 (100) | 8 (92) | 14 (94) | **17 (100)** |
| **XGBoost** | 3 | 128 | 4 (100) | 8 (97) | 128 (100) | **16 (97)** | 19 (99) | 28 (99) |

Table 2: Same as Table 1, but where $\epsilon$ is used as the cutoff value for ΔCB and ΔES.

## D.2  Detailed results

This section provides an in-depth breakdown of experiments presented in the body. Results for each problem are presented in the order they appeared in Table 1.

In each of the following table, we report statistics for each of the following metrics:

1. Succeeded: whether or not an $\epsilon$-optimal point was returned.
2. Terminated: whether or not the stopping rule kicked in prior to reaching upper limit $T$.
3. Stopping Time: the number of function evaluations requested.
4. Regret: latent values $\sup_{\boldsymbol{x} \in \mathcal{X}} f(\boldsymbol{x}) - f(\boldsymbol{s}_t)$ where $\boldsymbol{s}_t \in \arg\max_{\boldsymbol{x} \in \mathbf{X}_t} \mu_t(\boldsymbol{x})$ is a maximizer of the posterior mean at the time of stopping; reported in $\log_{10}$ scale.
5. Excess Regret: latent values $\sup_{\boldsymbol{x} \in \mathcal{X}} f(\boldsymbol{x}) - f(\boldsymbol{s}_t) + \epsilon$ for runs where regrets exceeded $\epsilon$; reported in $\log_{10}$ scale.

For the final three metrics, medians and interquartile ranges are shown alongside the mean. Similar to the preceding section, results for ΔCB and ΔES are reported using $\epsilon$ as the cutoff values to give a better picture of how these methods perform in the absence of post-hoc calibration.

| Metric | Stat. | **Oracle**[†] | **Budget**[†] | **Acq** | **ΔCB** | **ΔES** | **PRB** |
|---|---|---|---|---|---|---|---|
| **Succeeded** | mean | 1.00 | 0.96 | 1.00 | 0.89 | 0.94 | 0.97 |
| **Terminated** | mean | 1.00 | 1.00 | 1.00 | 1.00 | 1.00 | 1.00 |
| **Stopping Time** | mean | 10.52 | 17.00 | 30.14 | 13.02 | 15.01 | 17.81 |
| | 25% | 8.00 | 17.00 | 24.00 | 10.00 | 12.00 | 14.00 |
| | 50% | 9.00 | 17.00 | 28.50 | 12.00 | 14.00 | 17.00 |
| | 75% | 12.00 | 17.00 | 36.00 | 15.00 | 17.00 | 21.00 |
| **Regret** | mean | -5.71 | -3.59 | -4.78 | -2.49 | -3.07 | -3.47 |
| | 25% | -6.07 | -4.33 | -4.99 | -2.69 | -3.47 | -3.92 |
| | 50% | -5.36 | -3.20 | -4.36 | -2.04 | -2.57 | -2.99 |
| | 75% | -4.82 | -2.30 | -4.00 | -1.44 | -2.00 | -2.36 |
| **Excess Regret** | mean | – | -1.26 | – | -0.69 | -0.93 | -1.15 |
| | 25% | – | -1.32 | – | -1.00 | -1.10 | -1.30 |
| | 50% | – | -0.98 | – | -0.86 | -0.98 | -1.03 |
| | 75% | – | -0.92 | – | -0.21 | -0.87 | -0.94 |

Table 4: Results on GP[†] in $D = 2$ dimensions with noise variance $\gamma^2 = 10^{-6}$.

| Metric | Stat. | **Oracle**[†] | **Budget**[†] | **Acq** | **ΔCB** | **ΔES** | **PRB** |
|---|---|---|---|---|---|---|---|
| **Succeeded** | mean | 1.00 | 0.96 | 1.00 | 1.00 | 0.91 | 0.99 |
| **Terminated** | mean | 1.00 | 1.00 | 0.76 | 0.94 | 1.00 | 1.00 |
| **Stopping Time** | mean | 10.35 | 22.00 | 77.62 | 84.80 | 18.36 | 27.06 |
| | 25% | 8.00 | 22.00 | 43.75 | 70.00 | 14.75 | 17.00 |
| | 50% | 10.00 | 22.00 | 78.50 | 82.00 | 18.00 | 23.00 |
| | 75% | 12.00 | 22.00 | 126.25 | 99.00 | 21.00 | 32.00 |
| **Regret** | mean | -4.48 | -2.73 | -3.48 | -3.34 | -2.51 | -2.66 |
| | 25% | -4.55 | -2.63 | -3.56 | -3.61 | -2.40 | -2.65 |
| | 50% | -3.97 | -1.88 | -2.88 | -2.87 | -1.77 | -1.91 |
| | 75% | -3.28 | -1.46 | -2.31 | -2.30 | -1.32 | -1.59 |
| **Excess Regret** | mean | – | -1.59 | – | – | -1.54 | -0.11 |
| | 25% | – | -1.84 | – | – | -1.60 | -0.11 |
| | 50% | – | -1.66 | – | – | -1.21 | -0.11 |
| | 75% | – | -1.41 | – | – | -1.02 | -0.11 |

Table 6: Results on GP[†] in $D = 2$ dimensions with noise variance $\gamma^2 = 10^{-2}$.

| Metric | Stat. | **Oracle**[†] | **Budget**[†] | **Acq** | **ΔCB** | **ΔES** | **PRB** |
|---|---|---|---|---|---|---|---|
| **Succeeded** | mean | 1.00 | 0.95 | 1.00 | 0.66 | 0.74 | 0.99 |
| **Terminated** | mean | 1.00 | 1.00 | 0.96 | 1.00 | 1.00 | 0.99 |
| **Stopping Time** | mean | 28.48 | 64.00 | 86.36 | 25.61 | 30.36 | 63.68 |
| | 25% | 17.75 | 64.00 | 69.75 | 18.00 | 22.75 | 46.00 |
| | 50% | 26.00 | 64.00 | 90.00 | 23.00 | 28.00 | 64.00 |
| | 75% | 34.00 | 64.00 | 104.00 | 30.25 | 37.00 | 78.00 |
| **Regret** | mean | -3.86 | -3.06 | -3.47 | -1.53 | -1.89 | -3.15 |
| | 25% | -4.24 | -3.56 | -3.77 | -1.98 | -2.50 | -3.49 |
| | 50% | -3.73 | -3.11 | -3.41 | -1.49 | -1.84 | -3.10 |
| | 75% | -3.33 | -2.67 | -3.00 | -0.81 | -0.90 | -2.70 |
| **Excess Regret** | mean | – | -1.32 | – | -0.81 | -0.87 | -2.67 |
| | 25% | – | -1.70 | – | -1.26 | -1.25 | -2.67 |
| | 50% | – | -1.18 | – | -0.82 | -0.85 | -2.67 |
| | 75% | – | -0.62 | – | -0.47 | -0.53 | -2.67 |

Table 8: Results on GP[†] in $D = 4$ dimensions with noise variance $\gamma^2 = 10^{-6}$.

| Metric | Stat. | **Oracle**[†] | **Budget**[†] | **Acq** | **ΔCB** | **ΔES** | **PRB** |
|---|---|---|---|---|---|---|---|
| **Succeeded** | mean | 1.00 | 0.95 | 0.98 | 1.00 | 0.65 | 0.96 |
| **Terminated** | mean | 1.00 | 1.00 | 1.00 | 0.39 | 1.00 | 1.00 |
| **Stopping Time** | mean | 35.48 | 94.00 | 111.38 | 235.76 | 38.16 | 90.32 |
| | 25% | 19.00 | 94.00 | 87.75 | 223.75 | 26.00 | 67.00 |
| | 50% | 28.50 | 94.00 | 106.50 | 256.00 | 36.00 | 86.50 |
| | 75% | 42.00 | 94.00 | 138.25 | 256.00 | 46.00 | 113.25 |
| **Regret** | mean | -2.62 | -2.05 | -2.09 | -2.23 | -1.34 | -2.01 |
| | 25% | -2.66 | -2.35 | -2.35 | -2.46 | -1.66 | -2.21 |
| | 50% | -2.36 | -1.82 | -1.87 | -1.99 | -1.25 | -1.81 |
| | 75% | -1.92 | -1.52 | -1.62 | -1.73 | -0.87 | -1.50 |
| **Excess Regret** | mean | – | -1.50 | -1.72 | – | -1.08 | -1.57 |
| | 25% | – | -1.63 | -1.91 | – | -1.50 | -1.92 |
| | 50% | – | -1.45 | -1.72 | – | -0.95 | -1.49 |
| | 75% | – | -1.30 | -1.54 | – | -0.73 | -1.14 |

Table 10: Results on GP[†] in $D = 4$ dimensions with noise variance $\gamma^2 = 10^{-2}$.

| Metric | Stat. | **Oracle**[†] | **Budget**[†] | **Acq** | **ΔCB** | **ΔES** | **PRB** |
|---|---|---|---|---|---|---|---|
| **Succeeded** | mean | 0.99 | 0.95 | 0.98 | 0.50 | 0.65 | 0.98 |
| **Terminated** | mean | 0.99 | 1.00 | 0.92 | 1.00 | 1.00 | 0.96 |
| **Stopping Time** | mean | 55.33 | 124.00 | 150.91 | 35.82 | 48.86 | 141.24 |
| | 25% | 25.00 | 124.00 | 112.25 | 24.00 | 29.75 | 97.00 |
| | 50% | 39.50 | 124.00 | 142.50 | 31.00 | 45.50 | 133.50 |
| | 75% | 77.25 | 124.00 | 191.00 | 41.25 | 63.25 | 178.00 |
| **Regret** | mean | -3.21 | -2.60 | -2.85 | -1.11 | -1.52 | -2.79 |
| | 25% | -3.71 | -3.04 | -3.17 | -1.66 | -2.13 | -3.15 |
| | 50% | -3.15 | -2.69 | -2.78 | -1.03 | -1.67 | -2.77 |
| | 75% | -2.69 | -2.26 | -2.50 | -0.50 | -0.74 | -2.47 |
| **Excess Regret** | mean | -2.74 | -1.23 | -2.21 | -0.75 | -0.83 | -2.21 |
| | 25% | -2.74 | -1.54 | -2.47 | -1.04 | -1.11 | -2.47 |
| | 50% | -2.74 | -1.16 | -2.21 | -0.65 | -0.76 | -2.21 |
| | 75% | -2.74 | -0.84 | -1.94 | -0.23 | -0.30 | -1.94 |

Table 12: Results on GP[†] in $D = 6$ dimensions with noise variance $\gamma^2 = 10^{-6}$.

| Metric | Stat. | **Oracle**[†] | **Budget**[†] | **Acq** | **ΔCB** | **ΔES** | **PRB** |
|---|---|---|---|---|---|---|---|
| **Succeeded** | mean | 1.00 | 0.96 | 0.96 | 1.00 | 0.34 | 1.00 |
| **Terminated** | mean | 1.00 | 1.00 | 1.00 | 0.29 | 1.00 | 1.00 |
| **Stopping Time** | mean | 77.15 | 227.00 | 183.30 | 482.08 | 51.67 | 231.73 |
| | 25% | 33.00 | 227.00 | 138.00 | 499.00 | 30.50 | 170.00 |
| | 50% | 64.00 | 227.00 | 181.00 | 512.00 | 45.00 | 235.00 |
| | 75% | 96.50 | 227.00 | 219.00 | 512.00 | 63.50 | 295.00 |
| **Regret** | mean | -2.16 | -1.71 | -1.66 | -1.98 | -0.80 | -1.77 |
| | 25% | -2.41 | -1.97 | -1.85 | -2.25 | -1.19 | -2.02 |
| | 50% | -2.00 | -1.70 | -1.62 | -1.91 | -0.64 | -1.76 |
| | 75% | -1.72 | -1.37 | -1.35 | -1.60 | -0.38 | -1.53 |
| **Excess Regret** | mean | – | -1.51 | -1.98 | – | -0.72 | – |
| | 25% | – | -1.73 | -2.69 | – | -0.88 | – |
| | 50% | – | -1.60 | -2.13 | – | -0.56 | – |
| | 75% | – | -1.39 | -1.42 | – | -0.37 | – |

Table 14: Results on GP[†] in $D = 6$ dimensions with noise variance $\gamma^2 = 10^{-2}$.

| Metric | Stat. | **Oracle**[†] | **Budget**[†] | **Acq** | **ΔCB** | **ΔES** | **PRB** |
|---|---|---|---|---|---|---|---|
| **Succeeded** | mean | 1.00 | 0.95 | 1.00 | 0.30 | 0.41 | 0.88 |
| **Terminated** | mean | 1.00 | 1.00 | 0.92 | 1.00 | 1.00 | 0.98 |
| **Stopping Time** | mean | 38.75 | 79.00 | 87.94 | 19.41 | 24.25 | 62.57 |
| | 25% | 21.50 | 79.00 | 70.75 | 15.75 | 18.00 | 48.25 |
| | 50% | 34.00 | 79.00 | 91.50 | 18.00 | 22.00 | 61.00 |
| | 75% | 50.50 | 79.00 | 102.00 | 22.00 | 29.00 | 75.25 |
| **Regret** | mean | -3.80 | -3.08 | -3.41 | -0.70 | -1.01 | -2.70 |
| | 25% | -4.17 | -3.59 | -3.70 | -1.25 | -1.79 | -3.27 |
| | 50% | -3.66 | -3.13 | -3.27 | -0.34 | -0.59 | -2.85 |
| | 75% | -3.15 | -2.58 | -2.95 | -0.08 | -0.17 | -2.24 |
| **Excess Regret** | mean | – | -1.34 | – | -0.37 | -0.43 | -0.87 |
| | 25% | – | -1.70 | – | -0.53 | -0.59 | -1.41 |
| | 50% | – | -1.47 | – | -0.25 | -0.26 | -0.66 |
| | 75% | – | -0.88 | – | -0.04 | -0.07 | -0.49 |

Table 16: Results on GP in $D = 4$ dimensions with noise variance $\gamma^2 = 10^{-6}$.

| Metric | Stat. | **Oracle**[†] | **Budget**[†] | **Acq** | **ΔCB** | **ΔES** | **PRB** |
|---|---|---|---|---|---|---|---|
| **Succeeded** | mean | 1.00 | 0.95 | 0.97 | 0.80 | 0.22 | 0.92 |
| **Terminated** | mean | 1.00 | 1.00 | 0.99 | 0.60 | 1.00 | 0.99 |
| **Stopping Time** | mean | 58.53 | 157.00 | 130.53 | 179.00 | 28.60 | 100.40 |
| | 25% | 29.75 | 157.00 | 99.50 | 122.25 | 20.00 | 73.75 |
| | 50% | 50.00 | 157.00 | 128.50 | 224.00 | 27.00 | 100.50 |
| | 75% | 74.75 | 157.00 | 154.50 | 256.00 | 35.00 | 129.50 |
| **Regret** | mean | -2.59 | -2.03 | -2.05 | -1.78 | -0.52 | -1.82 |
| | 25% | -2.67 | -2.24 | -2.26 | -2.29 | -0.90 | -2.07 |
| | 50% | -2.32 | -1.79 | -1.79 | -1.76 | -0.33 | -1.69 |
| | 75% | -1.93 | -1.51 | -1.50 | -1.20 | -0.09 | -1.39 |
| **Excess Regret** | mean | – | -1.65 | -2.09 | -0.48 | -0.45 | -1.62 |
| | 25% | – | -2.83 | -2.40 | -0.84 | -0.74 | -1.78 |
| | 50% | – | -1.39 | -1.76 | -0.31 | -0.27 | -1.55 |
| | 75% | – | -0.63 | -1.62 | -0.01 | -0.06 | -1.31 |

Table 18: Results on GP in $D = 4$ dimensions with noise variance $\gamma^2 = 10^{-2}$.

| Metric | Stat. | **Oracle**[†] | **Budget**[†] | **Acq** | **ΔCB** | **ΔES** | **PRB** |
|---|---|---|---|---|---|---|---|
| **Succeeded** | mean | 1.00 | 0.95 | 1.00 | 0.99 | 1.00 | 0.99 |
| **Terminated** | mean | 1.00 | 1.00 | 0.00 | 1.00 | 1.00 | 1.00 |
| **Stopping Time** | mean | 17.40 | 25.00 | 64.00 | 29.39 | 31.27 | 32.17 |
| | 25% | 13.00 | 25.00 | 64.00 | 27.00 | 28.00 | 31.00 |
| | 50% | 18.00 | 25.00 | 64.00 | 31.00 | 32.00 | 33.00 |
| | 75% | 21.25 | 25.00 | 64.00 | 32.00 | 34.00 | 35.00 |
| **Regret** | mean | -6.27 | -2.14 | -5.93 | -2.79 | -3.00 | -3.10 |
| | 25% | -6.41 | -2.79 | -6.23 | -3.08 | -3.22 | -3.44 |
| | 50% | -6.34 | -1.97 | -5.99 | -2.75 | -2.95 | -3.07 |
| | 75% | -6.17 | -1.45 | -5.69 | -2.36 | -2.61 | -2.72 |
| **Excess Regret** | mean | – | -1.49 | – | 0.16 | – | 0.16 |
| | 25% | – | -1.62 | – | 0.16 | – | 0.16 |
| | 50% | – | -1.54 | – | 0.16 | – | 0.16 |
| | 75% | – | -1.15 | – | 0.16 | – | 0.16 |

Table 20: Results on Branin in $D = 2$ dimensions.

| Metric | Stat. | Oracle[†] | Budget[†] | Acq | $\Delta$CB | $\Delta$ES | PRB |
|---|---|---|---|---|---|---|---|
| **Succeeded** | mean | 1.00 | 0.96 | 1.00 | 0.83 | 0.84 | 1.00 |
| **Terminated** | mean | 1.00 | 1.00 | 1.00 | 1.00 | 1.00 | 1.00 |
| **Stopping Time** | mean | 13.89 | 22.00 | 29.00 | 15.06 | 16.93 | 21.34 |
| | 25% | 11.00 | 22.00 | 24.00 | 14.00 | 16.00 | 17.00 |
| | 50% | 13.00 | 22.00 | 26.00 | 15.00 | 17.00 | 19.00 |
| | 75% | 15.25 | 22.00 | 30.00 | 16.00 | 18.00 | 21.00 |
| **Regret** | mean | -6.59 | -3.59 | -4.71 | -1.78 | -2.53 | -3.39 |
| | 25% | -6.66 | -4.39 | -5.06 | -2.20 | -3.32 | -3.81 |
| | 50% | -6.64 | -3.81 | -4.63 | -1.98 | -2.77 | -3.41 |
| | 75% | -6.58 | -3.32 | -4.23 | -1.61 | -2.38 | -2.90 |
| **Excess Regret** | mean | – | -0.61 | – | -0.26 | -0.08 | – |
| | 25% | – | -0.80 | – | -0.17 | -0.17 | – |
| | 50% | – | -0.37 | – | -0.17 | -0.17 | – |
| | 75% | – | -0.18 | – | -0.17 | -0.17 | – |

Table 22: Results on Hartmann in $D = 3$ dimensions.

| Metric | Stat. | Oracle[†] | Budget[†] | Acq | $\Delta$CB | $\Delta$ES | PRB |
|---|---|---|---|---|---|---|---|
| **Succeeded** | mean | 0.67 | 0.67 | 0.67 | 0.46 | 0.56 | 0.64 |
| **Terminated** | mean | 0.67 | 0.00 | 1.00 | 1.00 | 1.00 | 1.00 |
| **Stopping Time** | mean | 104.85 | 256.00 | 41.30 | 23.15 | 29.43 | 40.52 |
| | 25% | 25.75 | 256.00 | 36.75 | 18.00 | 27.00 | 37.00 |
| | 50% | 35.00 | 256.00 | 39.50 | 26.00 | 30.00 | 40.00 |
| | 75% | 256.00 | 256.00 | 42.00 | 31.00 | 35.00 | 43.00 |
| **Regret** | mean | -4.12 | -4.12 | -2.44 | -0.91 | -1.56 | -2.30 |
| | 25% | -5.70 | -5.70 | -3.31 | -1.75 | -2.48 | -3.29 |
| | 50% | -5.69 | -5.69 | -2.88 | -0.85 | -1.98 | -2.82 |
| | 75% | -0.92 | -0.92 | -0.92 | 0.03 | -0.84 | -0.92 |
| **Excess Regret** | mean | -1.72 | -1.72 | -1.70 | -0.36 | -0.76 | -1.50 |
| | 25% | -1.72 | -1.72 | -1.71 | -1.09 | -1.52 | -1.71 |
| | 50% | -1.72 | -1.72 | -1.71 | -0.59 | -1.13 | -1.70 |
| | 75% | -1.72 | -1.72 | -1.69 | 0.47 | 0.42 | -1.67 |

Table 24: Results on Hartmann in $D = 6$ dimensions.

| Metric | Stat. | Oracle[†] | Budget[†] | Acq | $\Delta$CB | $\Delta$ES | PRB |
|---|---|---|---|---|---|---|---|
| **Succeeded** | mean | 1.00 | 0.95 | 1.00 | 0.99 | 1.00 | 1.00 |
| **Terminated** | mean | 1.00 | 1.00 | 1.00 | 1.00 | 1.00 | 1.00 |
| **Stopping Time** | mean | 32.78 | 46.00 | 95.28 | 67.05 | 71.66 | 84.17 |
| | 25% | 26.00 | 46.00 | 92.00 | 66.00 | 69.00 | 81.00 |
| | 50% | 33.00 | 46.00 | 95.00 | 68.00 | 71.00 | 84.00 |
| | 75% | 41.00 | 46.00 | 99.00 | 71.00 | 74.00 | 88.00 |
| **Regret** | mean | -9.00 | -4.93 | -9.00 | -8.38 | -8.85 | -9.00 |
| | 25% | -9.00 | -5.00 | -9.00 | -9.00 | -9.00 | -9.00 |
| | 50% | -9.00 | -4.60 | -9.00 | -9.00 | -9.00 | -9.00 |
| | 75% | -9.00 | -4.28 | -9.00 | -9.00 | -9.00 | -9.00 |
| **Excess Regret** | mean | – | -5.28 | – | -3.27 | – | – |
| | 25% | – | -5.35 | – | -3.27 | – | – |
| | 50% | – | -5.23 | – | -3.27 | – | – |
| | 75% | – | -4.60 | – | -3.27 | – | – |

Table 26: Results on Rosenbrock in $D = 4$ dimensions.

| Metric | Stat. | Oracle[†] | Budget[†] | Acq | $\Delta$CB | $\Delta$ES | PRB |
|---|---|---|---|---|---|---|---|
| **Succeeded** | mean | 1.00 | 0.96 | 1.00 | 0.92 | 0.94 | 1.00 |
| **Terminated** | mean | 1.00 | 1.00 | 0.00 | 1.00 | 0.98 | 0.96 |
| **Stopping Time** | mean | 4.98 | 11.00 | 64.00 | 11.55 | 20.37 | 24.30 |
| | 25% | 2.00 | 11.00 | 64.00 | 6.00 | 11.00 | 10.25 |
| | 50% | 4.00 | 11.00 | 64.00 | 8.00 | 14.00 | 17.00 |
| | 75% | 7.00 | 11.00 | 64.00 | 16.00 | 26.00 | 33.25 |
| **Regret** | mean | -5.05 | -2.89 | -4.22 | -2.75 | -3.18 | -3.47 |
| | 25% | -9.00 | -2.92 | -3.52 | -2.92 | -3.15 | -3.28 |
| | 50% | -3.52 | -2.80 | -3.30 | -2.72 | -2.96 | -3.05 |
| | 75% | -3.40 | -2.59 | -3.15 | -2.55 | -2.74 | -2.83 |
| **Excess Regret** | mean | – | -2.32 | – | -2.68 | -2.87 | – |
| | 25% | – | -2.39 | – | -2.93 | -3.34 | – |
| | 50% | – | -2.32 | – | -2.66 | -2.68 | – |
| | 75% | – | -2.24 | – | -2.42 | -2.31 | – |

Table 28: Results on CNN in $D = 4$ dimensions.

| Metric | Stat. | Oracle[†] | Budget[†] | Acq | $\Delta$CB | $\Delta$ES | PRB |
|---|---|---|---|---|---|---|---|
| **Succeeded** | mean | 1.00 | 0.97 | 1.00 | 0.97 | 0.99 | 0.99 |
| **Terminated** | mean | 1.00 | 1.00 | 0.21 | 1.00 | 1.00 | 1.00 |
| **Stopping Time** | mean | 3.74 | 8.00 | 121.73 | 16.90 | 19.57 | 28.51 |
| | 25% | 2.00 | 8.00 | 128.00 | 11.00 | 13.00 | 22.50 |
| | 50% | 3.00 | 8.00 | 128.00 | 16.00 | 19.00 | 28.00 |
| | 75% | 6.00 | 8.00 | 128.00 | 21.00 | 23.75 | 34.50 |
| **Regret** | mean | -8.52 | -2.83 | -3.79 | -3.25 | -3.34 | -3.59 |
| | 25% | -9.00 | -2.82 | -3.61 | -3.21 | -3.31 | -3.31 |
| | 50% | -9.00 | -2.66 | -3.31 | -2.83 | -2.87 | -3.07 |
| | 75% | -9.00 | -2.48 | -3.01 | -2.66 | -2.62 | -2.83 |
| **Excess Regret** | mean | – | -2.06 | – | -2.39 | -2.64 | -2.64 |
| | 25% | – | -2.19 | – | -2.69 | -2.64 | -2.64 |
| | 50% | – | -1.94 | – | -2.64 | -2.64 | -2.64 |
| | 75% | – | -1.86 | – | -2.21 | -2.64 | -2.64 |

Table 30: Results on XGBoost in $D = 3$ dimensions.

