# OpenReview forum: "Stopping Bayesian Optimization with Probabilistic Regret Bounds"
_NeurIPS.cc/2024/Conference — NeurIPS 2024 poster_

### Official Review · Reviewer_cuEF · 2024-07-07

**Soundness:** 2
**Presentation:** 1
**Contribution:** 2
**Rating:** 4
**Confidence:** 3

**Summary:**

The paper introduces a Monte-Carlo-based stopping criterion for Bayesian optimization tasks, which has been neglected in the Bayesian optimization literatures. The authors provide a generic algorithm that can be tailored to Bayesian optimization setting and establish theoretical results and demonstrate numerical performances.

**Strengths:**

The paper provides many interesting theoretical tools with relevant references that could be potentially useful in establishing stopping rules and other understandings in Bayesian optimization tasks.

**Weaknesses:**

The paper's writing can be improved significantly. Although the paper introduces the most generic version of the algorithm as in Algorithm 1, to be actually used in the BO setting, one needs to calibrate parameters $(\delta^t_{est})_{t\in \mathbb{N}}$ that are highly nontrivial. Instead of presenting algorithm 1, readers would benefit more by seeing the algorithm directly targeted to the Bayesian optimization setting. Or perhaps it would be better if authors title the paper more broadly and consider applications beyond the Bayesian optimization. Section 3.2, in particular, needs to be modified to clarify its connection to the Bayesian optimization setting. A proper description of parameter scheduling and end-to-end algorithmic description would make the paper more solid and appreciated by a wider audience.

**Questions:**

1. The regret defined in (2) is not a simple regret, and it does not accurately measure the performance of the optimization as the objective function $f$ is not used, but instead a sequence of objective functions $(f_t)_{t \in \mathbb{N}}$, which vary over time, is considered. Can you clarify what the regret means in this case? It seems it makes more sense to stop the BO algorithm when $f(x^*) - f(x_t) \le \epsilon$ with probability at least $1-\delta$. In the later parts of the manuscript, $f_t$ seems to be a sample path, and if this is the case, algorithms like GP-TS would easily satisfy the condition, which authors refer to as probabilistic regret bound, in the early stage of the algorithm. Such a condition would be virtually non-informative.

2. In line 110, the authors claim that for each draw of $f_t$, the indicator can be evaluated using gradients. Can you elaborate more on this?

**Limitations:**

Not applicable.

---

> ### Author Rebuttal · Authors · 2024-08-06
>
> Thank you for reviewing our submission. We are sorry to hear that you found our presentation of the material underwhelming. Some of your feedback was similar to that of another reviewer, so please see our global response for additional details.
>
>
> **What does $f_t$ represent?**
>
> Per the Section 2, $f_t$ denotes the posterior at time $t$ and this remains the case throughout the paper. In Section 3.1, we do talk about generating sample paths, but we usually say things like "draws of $f_t$". The only exception we have found is on line 102 where it says "drawing functions $f_t$” instead of "generating function draws of $f_t$". We will amend this line (and any others) to avoid confusion.
>
> This also addresses your concern that algorithms like GP-TS would easily satisfy the proposed criterion. This is not the case because we use Monte Carlo methods to integrate out $f_t$ when evaluating (3).
>
>
> **$r_t(x)$ is not a simple regret**
>
> Equation (2) defines the model-based (simple) regret as $r_t(x) = f_t^* - f_t(x)$, where $f_t^* = \sup_{x \in \mathcal{X}} f_t^*$. Like $f_t$, $r_t(x)$ is a random variable. It denotes our regret for taking $x$ as our solution _according to the model_. Said differently, if we believe that the true function was drawn from the prior and that observations are indeed corrupted by independent Gaussian noise with variance $\gamma^2$, then $P(r_t(x) < \epsilon)$ is the subjective probability that $x$ satisfies the criterion given what is known a time $t$.
>
> Hence, the proposed stopping rule does exactly what you suggested: it stops when the regret is less than $\epsilon$ with probability at least $1 - \delta$. The only difference is that the regret in question is defined under the model because the true supremum (hence the true regret) is unknown.
>
>
> **Scheduling $\delta_{\text{est}}^t$**
>
> We agree that additional details are needed here and will include them in future versions. In our experiments, we set $\delta_{\text{est}}^t = T^{-1}$ where $T$ was an upper bound on the number of queries (line 245). As seen in Figure 3, the number of samples requested by Algorithm 1 increased very slowly in $\delta_{\text{est}}$, so this upper bound can be loose (e.g. hundreds of times larger than the anticipated number of queries). As will be shown in future versions, this trend holds for different approaches to bounding estimation errors, such as Clopper-Pearson intervals and equal-tailed Bayesian credible intervals with uninformative priors.
>
> In cases where no such upper bound is available, we recommend using an "outer schedule" $(\delta_{\text{est}}^t)$ that is similar to the "inner schedule" for $(n_j)$ used by Algorithm 1, which is discussed in Appendix B.1. Under this approach, the stopping criterion is only evaluated at steps $\beta^i t_0$ for $i \in \mathbb{N}$, where $t_0 \in \mathbb{N}$ is an initial time and $\beta > 1$ is a scheduling parameter. The choice of $\beta$ will depend on the ratio of the costs of making a query versus the cost of evaluating the stopping rule (approaching one as the ratio goes to infinity). Relative to the convergence time $T_0$ in Proposition 1 (and $S$ in the proof of Proposition 2), this approach results in at most $\beta$ times more queries.

---

> > ### Comment · Reviewer_cuEF · 2024-08-11
> > **Response to the authors**
> >
> > The usage does not really convince me of the model-based regret. Although Bayesian optimization algorithms endow probabilistic structure to the objective function, it often just serves as an intermediate assumption to derive an algorithm. The algorithm itself has a strong performance guarantee under model misspecification (e.g., objective function in the RKHS). If one must use such model-based regret, this paper needs to emphasize the importance of correct model specification and explicitly state the limitation in the misspecified model setup. In practice, there are a substantial number of successful BO examples with model misspecifications. I don't find then the framework provided in this paper attractive or practical. Therefore, I will retain my score.

---

> > > ### Author Response · Authors · 2024-08-11
> > > **Regarding model mismatch**
> > >
> > > We are surprised to see that your review has pivoted to focusing on model misspecification and request you discuss this matter with your fellow reviewers. Our (potentially biased) view is that we have been very upfront about this issue. Here is some evidence to support this claim:
> > > - We state in the abstract that our goal is to find an $\epsilon$-optimal point with probability at least $1 - \delta$ _under the model_.
> > >
> > > - We reinforce this point in the introduction by discussing how model misspecification is one of two key issues that has likely impeded the development of model-based stopping rules; and, we explicitly tell the reader on line 31 that our work only provides "mild commentary" on this topic.
> > > - We qualify statments by saying "under the model" or "according to the model" more than 10 times in the main text.
> > > - We refer to (2) as the "model-based regret" on line 81.
> > > - The explanation of our main convergence result, Proposition 2, says: "If the model is correct, we can therefore promise to return a satisfactory solution with high probability."
> > >
> > > - Model mismatch is the focus of Section 5.3.
> > >
> > > - In our conclusion, we say: "If data is generated according to the model, we can therefore guarantee that BO is likely to return a satisfactory solution."
> > >
> > > To be clear, our intention is not to say that your dissatisfaction is unwarranted. We fully agree that model misspecification is a major issue. At the same time, we argue that we (as a community) can only make progress on model-based stopping if we do so incrementally. To our knowledge, our contributions are more than incremental:
> > > - We are the first to provide actual performance guarantees (even if they are defined under a model).
> > > - Our convergence results are much more general than any preceding work.
> > > - We demonstrate significantly superior empirical performance.
> > >
> > > If you disagree, then please explain why or point to related works that have done a better job.

---

> > > > ### Comment · Reviewer_cuEF · 2024-08-11
> > > > **On model misspecification**
> > > >
> > > > Based on my understanding of your previous response, the main concern I have is the utility of the algorithm which relies on unrealistically strong assumptions. To be clearer, 1) your stopping rule only makes sense, when the objective function is a realization of the posterior Gaussian process, however, many numerical examples you provide, e.g., Branin, Hartman, Rosenbrock fall into the RKHS assumption setting. In theory, these functions are too smooth to be considered to be a sample path of the posterior Gaussian process. In practice, unless, you run the algorithm long enough, your draws of the posterior Gaussian process cannot even approximate these objective functions. In addition, 2) your stopping rule is based on the posterior Gaussian process, and these posteriors can be significantly different from the objective functions with a few data points, as mentioned before. Then the stopping criterion with such a misspecified posterior GP is completely meaningless. How can you make sure that the objective function is at least somewhat close to some sample path of the posterior GP? Unless there is guidance on this, I am not convinced how to utilize this stopping criterion. In a nutshell, although I am not opposed to developing a model-based stopping rule, it must reflect practical Bayesian optimization applications and the current methodology doesn't seem to do so.

---

> > > > > ### Author Response · Authors · 2024-08-12
> > > > > **Re: on model misspecification**
> > > > >
> > > > > Thank you for your continued input!
> > > > >
> > > > > > your stopping rule only makes sense, when the objective function is a realization of the posterior Gaussian process, however, many numerical examples you provide, e.g., Branin, Hartman, Rosenbrock fall into the RKHS assumption setting. In theory, these functions are too smooth to be considered to be a sample path of the posterior Gaussian process. In practice, unless, you run the algorithm long enough, your draws of the posterior Gaussian process cannot even approximate these objective functions.
> > > > >
> > > > > Our method only requires that uncertainty for the true function becomes sufficiently small. If the true function is smoother than the GP sample paths, then our method may be conservative. Our analysis still shows that the proposed stopping rule will converge however (under appropriate assumptions). Further, our applied results suggest that it does so efficiently.
> > > > >
> > > > > > your stopping rule is based on the posterior Gaussian process, and these posteriors can be significantly different from the objective functions with a few data points, as mentioned before. Then the stopping criterion with such a misspecified posterior GP is completely meaningless. How can you make sure that the objective function is at least somewhat close to some sample path of the posterior GP? Unless there is guidance on this, I am not convinced how to utilize this stopping criterion.
> > > > >
> > > > > To our knowledge, there is no way of avoiding this. If we compare with the frequentist/bandit setting where one assumes that $f$ has bounded RKHS norm $B$, then the same issue emerges if we underestimate $B$. There is simply no free lunch.

---

> ### Comment · Reviewer_cuEF · 2024-08-12
> **Re: Re: on model misspecification**
>
> Thanks for the comment as well.
>
> I think this paper has many interesting technical works that could be potentially useful in developing stopping rules for Bayesian optimization. However, to make my stance clearer, what I find more appropriate is to make sure the stopping criterion reflects how close the output of the algorithm is to the optimum of the objective function. As the authors clarified in the response, the stopping criterion would be appropriate without model misspecification, however, since a substantial number of BO applications fall into the misspecified model setting, if one develops a model-based stopping criterion, one must also provide a diagnostics/heuristics to guide practitioners to safely use tools. Currently, I find descriptions in Section 5.3 are insufficient. If the authors are willing to provide more down-to-earth guidance on the choice of a number of iterations to make sure the posterior Gaussian process can provide a reasonable stopping criterion for the deterministic black-box optimization in the manuscript, I am willing to raise the score a bit.

---

> > ### Author Response · Authors · 2024-08-12
> > **Re^3: model misspecification**
> >
> > Like what you've said, we view our work as more of a step in the right direction than a final solution. It is vital that readers understand both the strength and weaknesses of methods such as ours in order to make informed decisions. Thank you for stressing this point. We will provide extended discussion of this topic regardless of whether or not you choose to adjust your score.

---

> > > ### Comment · Reviewer_cuEF · 2024-08-12
> > > **Re^4**
> > >
> > > Since technical misunderstandings on the regret have been resolved, I changed the score.
> > >
> > > However, I think the current work is still yet to be considered for publication unless there is more concrete guidance on bridging model-based regret with the actual discrepancy.

---

### Official Review · Reviewer_GLW6 · 2024-07-10

**Soundness:** 3
**Presentation:** 3
**Contribution:** 2
**Rating:** 5
**Confidence:** 3

**Summary:**

This paper develops a $(\varepsilon,\delta)$ stopping criterion for Bayesian optimization algorithms.  The authors propose a probabilistic regret bound estimator, which is constructed through sampling the function and find the maximum points of these samples,  to decide when to stop the algorithm. They also give the theoretical analysis of this estimator and show their results in the numerical experiments.

**Strengths:**

* The paper is well written and well organized.
* The problem formulation is clear.
* The algorithms are supported by some theoretical results (though I think the assumptions have some problem).

**Weaknesses:**

My concern is about the Assumption A.3, which is not a common assumption in BO field. This assumption, as you mentioned, is true for a generalized EI in [1]. However, it is still unknown if this assumption is held in a noisy case, and I am also not agree with that the Knowledge gradient is in the form of generalized EI defined in [1]. I think the UCB algorithm with a fix coefficient will never satisfiy this assumption. As a result, the theoretical analysis is restricitive and only suitable for several specific acquisition functions.




[1] Convergence properties of the expected improvement algorithm with fixed mean and covariance functions.

**Questions:**

* I didn't understand the right figure in Fig 2. Do you mean that you change the posterior distribution for orange, green and red line through conditioning on the partial information of real optima?

* How to decide the oracle budget in the experiment section? As the budget is chosen by an oracle, why sometimes the budget may be worse than other methods?

**Limitations:**

They adequately addressed the limitations.

---

> ### Author Rebuttal · Authors · 2024-08-06
>
> Thank you for your feedback. We will respond to each of your comments below.
>
>
> **Assumption A3**
>
> In order to make high-probability statements about $f$'s global properties, we must eventually learn enough about the function. We chose to enforce this condition by assuming that $(x_t)$ is dense in $\mathcal{X}$ because it is simple to understand and popular strategies exhibit this behavior (in the noise-free setting). This condition can also be guaranteed by using an $\epsilon$-greedy strategy. We argue this practice is desirable because it helps protect against model mismatch and ensures asymptotic performance no worse than that of random search.
>
> This being a paper on Bayesian optimization however, that conclusion is somewhat underwhelming. As you’ve said, the set of acquisition functions (AFs) that satisfy A3 is an open question (even when $f$ is noise-free). Consider the following analysis.
>
> Let $s_t \in \arg\max_{x \in \mathcal{X}} \mu_t(x)$ and $x_t \in \arg\max_{x \in \mathcal{X}} \alpha_t(x)$, where $\alpha_t$ denotes the chosen AF. The proof strategy from [1] is then:
> 1. Define $\alpha_t$ as a non-negative, continuous function such that $\alpha_t(x) = 0 \iff \sigma_t(x) = 0 \land \mu_t(x) - \mu_t(s_t) \le 0$.
> 2. Show that  $\sigma_t(x_t)$ and $\mu_t(x_x) - \mu_t(s_t)$ are both non-positive when $t \to \infty$.
> 3. Show that $\alpha_t(x_t)$ therefore vanishes as $t \to \infty$.
> 4. Show that (3), together with the no-empty-ball property [1] and the definition of $x_t$, implies that $(x_t)$ is dense in $\mathcal{X}$.
>
> Notice that the proof is actually simpler if $\alpha_t(x) = 0 \iff \sigma_t(x) = 0$, i.e. if a query has no value if and only if it provides no information. This family of AFs not only contains KG but other common non-myopic AFs (such as variants of entropy search). We will double check this result and update the paper accordingly. Note that the statement in the paper is not quite correct anyways since assumption A2 does not guarantee the NEB property.
>
> If you agree with this analysis, then we hope you will reconsider your position on A3. Requiring noise-free assumption is undesirable, but this topic is beyond the scope of our submission. To our knowledge, previous works have made far stronger assumptions (such as use of GP-UCB with the theoretically correct $\beta_t$) or omitted proofs entirely.
>
> p.s. We also think that UCB with fixed $\beta_t$ will not be space filling. See Section 5 of [2].
>
>
> **Figure 2**
>
> We will improve the related text, but your interpretation is correct. For instance, $f_t(x) | f_t(x) \le f_t^*(x)$ in green means that we truncated prior to approximating (3), i.e. we sampled $f_t^*$ and averaged over $P(f_t^* - f_t(x) \le \epsilon \mid f_t(x) \le f_t^*)$.
>
> Properly conditioning on $f_t^*$ would require us to truncate everywhere and marginalize over random sets $\arg\max f$, which is intractable. To illustrate this, let $f_1 \sim N(0, 1)$ and $f_2 \sim N(0, 10^{-9})$ be independent and suppose you are told that $\max(f_1, f_2) = 3$. How confident can you be that $f_1 = 3$?
>
> We ultimately ended up sampling functions $f_t$ instead, which avoids this issue. We thought it was interesting to show the impact of different ways of communicating information about the supremum. In hindsight, it may be better to move this to appendices.
>
>
> **Oracle budgets**
>
>  Oracle budgets were defined retroactively by looking at the results and choosing, for each problem, the smallest time $T$ so that $95\%$ of runs met the regret goal (where possible). Table 1 reports medians. If 94 out of 100 runs on a hypothetical problem took 10 trials to satisfy the regret bound and the remaining 6 took 50, then the oracle budget (and hence the median) would be 50. The median stopping times of other methods might be closer to 10 however. We will include problem-specific result tables in extended results that give a more detailed breakdown of our results.
>
> #### References
> [1] Vazquez and Bect, 2010. "Convergence properties of the expected improvement algorithm with fixed mean and covariance functions".
>
> [2] Jones, 2001. "A Taxonomy of Global Optimization Methods Based on Response Surfaces".

---

> > ### Comment · Reviewer_GLW6 · 2024-08-11
> >
> > I'd like to thank the authors for their response and for clarifying several issues.
> >
> > I think the analysis process you provided is correct, but it still cannot reduce my doubts about the theoretical part. There are no assumptions about Aq in the algorithm description, but there are many limitations in the theoretical section, which makes the algorithm and theory in the article inconsistent. This paper still has potential for improvements such as providing some experiments about the stopping criterion on different Aq, so I will not change the score.

---

> ### Author Response · Authors · 2024-08-12
>
> Thank you for your input! Please humor a quick reply.
>
> In the noise-free setting, we believe to have shown (either in the text or in follow-up discussion here) that PRB converges for most acquisition functions (AFs). The only well-know exception we are aware of is UCB with a fixed $\beta$ parameter. At the risk of being overly pedantic: this is technically a bandit method and therefore falls outside the scope our work.
>
> Regarding inconsistencies, assumptions A1-A3 could be communicated earlier in the text. We have not done so because they are only required for our technical analysis. Our algorithm can be used with any search strategy, but convergence can only be guaranteed if additional assumptions are made. This is unavoidable for stopping rules like PRB, since we need to protect against degenerate cases where, e.g., only a single point is ever queried.
>
> Finally, we agree with your comment that we should show results for more than one AF. This is a good suggestion. We actually did run some experiments early on with different AFs for standard problem like Branin; and, these were consistent with results reporting in the text. We did/do not have the compute resources to run all of our experiments with multiple AFs, but are happy to add experiments that illustrate how our method interacts with different strategies.
>
> We ask you to reconsider not to revising your score, but respect your decision either way.

---

### Official Review · Reviewer_pJBb · 2024-07-11

**Soundness:** 3
**Presentation:** 3
**Contribution:** 3
**Rating:** 7
**Confidence:** 3

**Summary:**

This paper addresses the challenge of determining when to stop a Bayesian optimization process. Traditional methods rely on exhausting a predefined budget, but this work proposes an alternative approach based on probabilistic criteria as a stopping criterion.
Key Contributions:
New Stopping Criterion: The paper introduces a stopping criterion based on the probability that a solution is within a certain threshold ($\epsilon$) of the optimal solution with high confidence (1 - $\delta$). This criterion is more adaptive and user-friendly compared to traditional budget-based stopping rules.
Theoretical Guarantees: The authors prove that Bayesian optimization satisfies this new stopping criterion under mild technical assumptions for Gaussian process priors.
Practical Algorithm: This paper presents a practical algorithm for evaluating the stopping rules using Monte Carlo estimators. This algorithm is designed to be robust against estimation errors.
The paper provides empirical results demonstrating the effectiveness and limitations of the proposed approach.

**Strengths:**

This paper introduces a new stopping criterion for Bayesian optimization based on probabilistic regret bounds, addressing the challenge of indeterminate budget allocation in black-box optimization scenarios. The criterion is formulated as $P(r(x*) \le \epsilon) \ge 1 - \delta$  where $r(x*)$ represents the regret of the best solution found, $\epsilon$ is the acceptable regret threshold, and $\delta$ is the confidence parameter. This probabilistic approach provides a more adaptive and theoretically grounded alternative to traditional fixed-budget stopping rules, allowing the optimization process to terminate when a solution is found that is likely to be within $\epsilon$ of the global optimum with high confidence $1-\delta$

**Weaknesses:**

N/A

**Questions:**

Assumption 3 (A3), which requires the sequence of query locations $(\vec{x}_t)$ to be almost surely dense in the search space $\mathcal{X}$. Is this assumption practical in the high-dimensional data ?

---

> ### Author Rebuttal · Authors · 2024-08-04
>
> We are pleased to hear that you enjoyed our submission.
>
>
> **Is A3 practical for high-dimensional problems?**
>
> We give a detailed reply below. The short answer is that what matters in practice is how smooth $f$ is relative to the size of $\mathcal{X}$.
>
>
> For simplicity, assume $f \sim \mathcal{GP}(0, k)$ where $k$ is a stationary kernel with unit variance. In discussing dimensionality, we are really talking distances. We used the infinity norm to measure distance in the main text for convenience. From a theoretical perspective, the particular choice of norm isn't so important here: $\mathcal{X}$ is assumed finite-dimensional, so all norms are equivalent. What ultimately matters is the distance induced by $k$. This is easy to see in the sense that even if $||x - x'||$ is very large, the distance passed to the kernel may be very small (i.e., $f$ may be very smooth). Even this distance can be misleading however; consider a periodic kernel.
>
> It turns out that the right distance for characterizing the supremum is the canonical pseudo-metric (29)
>
> $
> d_k(x, x') = \sqrt{k(x, x) - 2 k(x, x') + k(x' , x')}.
> $
>
> To build intuition for why this is the appropriate metric, notice that $d_k(x, x')$ goes from zero to four as the correlation between $f(x)$ and $f(x')$ goes from positive to negative one. Denoting this correlation by $\rho$,  one can show that
>
> $
> \mathbb{E}\left[\max(f(x), f(x'))\right] = \sqrt{\frac{1 - \rho}{\pi}},
> $
>
> which exhibits the same trend. Further details can be found in Appendix A.

---

> > ### Comment · Reviewer_pJBb · 2024-08-11
> >
> > Thank you for providing a detailed response to my question. I appreciate the clarification regarding the practicality of Assumption 3 (A3) for high-dimensional problems. Your explanation about the role of smoothness of the function $f$ relative to the size of  $\mathcal{X}$ is insightful. The discussion on the choice of norm and its equivalence in finite-dimensional spaces, as well as the importance of the kernel-induced distance, is particularly helpful.

---

### Official Review · Reviewer_5ozp · 2024-07-15

**Soundness:** 3
**Presentation:** 3
**Contribution:** 3
**Rating:** 6
**Confidence:** 4

**Summary:**

This paper proposes estimators for Bayesian optimisation stopping rules backed by theoretical guarantees. The stopping criterion estimators are probably approximately correct (PAC) and derived from sample-based estimates of algorithm’s simple regret according to a Gaussian process model. Experiments complement the theoretical findings on benchmark problems involving synthetic-data benchmarks and hyper-parameter tuning problems.

**Strengths:**

* The text is mostly well written.
* The theoretical results seem rigorous and the involved analysis techniques might benefit other areas of research in BO and bandits.
* A reasonable number of baselines and benchmarks is included in the experiments and significant improvements are shown for the proposed approach.

**Weaknesses:**

* Some important details of the methodology are not clear, as the description tends to be quite verbose with a lack of technical details. It is not very clear how Algorithm 1 applied within the BO loop, for example.
* Scalability limitations are not discussed.

**Questions:**

* In Algorithm 1, what exactly is $Z$, given the dependence of $\Psi$ on $\mathbf{x}$?
* It’s not clear how gradients are applied to evaluate the regret-bound indicators, as suggested in line 110. The sampled functions are likely to be non-convex. So, even if running gradient descent from the given $\mathbf{x}$ does not lead to a point $\mathbf{x’}$ satisfying $f_t(\mathbf{x’}) - f_t(\mathbf{x}) > \epsilon$, there’s no guarantee that elsewhere on the search space this would not be achieved.
* What was the runtime for the algorithms in the experiments reported in Table 1? Since the PRB strategy requires multiple samples from the GP at each iteration, it seems to me that the cost of running the algorithm ramps up quite fast. So it’d be good to compare the PRB runtimes against the baselines’ to have an idea of how much more computational resources are required by attempting an early stopping strategy.
* What was the feature map $\boldsymbol{\phi}$ used for the experiments?
* What is $D$ in the definition of $\beta_t$ in line 270?
* Hartmann, Rosenbrock and Branin are classic synthetic test functions for optimisation algorithms. So why are they included under a section labeled “Results on real problems”? That can be somewhat misleading, as we’d usually refer to “real problems” as cases involving real data or challenging application scenarios attached to a real-world application (e.g., complex physics simulators for scientific problems).
* Scalability limitations are not discussed. For instance, the proposed PRB estimator is based on optimisation over samples from a GP, and finding their optima should become harder and harder as the dimensionality of the space grows. Are there useful approximations that could be applied without hurting the theoretical guarantees by too much? Also, the algorithm could be adapted to work with sparse GP models, especially sparse spectrum GPs, given the feature-based approximations for sampling in Eq. 4. However, I’ve missed a discussion on how to adapt the algorithm to scenarios involving large amounts of data (e.g., batch BO).

**Limitations:**

Some limitations are discussed in the main text, while other issues, such as scalability, might not have been addressed, as mentioned above.

---

> ### Author Rebuttal · Authors · 2024-08-04
>
> Thank you for your detailed feedback. We will address most of your points below.
>
>
> **Missing details**
>
> Your comment that important details about how the presented material is used in the context of BO has been heard. We will rework the text to ensure this content is clearly communicated. Please see our global response for further details on this topic.
>
>
> **Scalability**
>
> This is an interesting direction for future work. We agree that works about stopping criteria would ideally address the scaling issues they may invite. At the same time, we believe to have provided first-of-their-kind results for model-based stopping of BO. Further, we have taken the time to study how many theoretical model-based stopping algorithms can be converted to robust, practical ones. You have said that these results not only seem rigorous but (potentially) beneficial to related fields.  We therefore think it fair to say this topic is beyond the scope of our work.
>
>
> **Questions**
> 1. The random variable $Z$ is introduced on line 128 and its definition in the specific context of PRB is given on line (134). In writing Section 3.2, we wanted to emphasize the fact that these methods are generic. Based on the feedback we received however, it seems that this ended up being confusing.
>
> 1. What you have said is correct. Please see our global response for discussion of how we optimized $f_t$.
>
> 1. Runtimes are shown on the right in Figure 3. We expressed these as a function of the distance between the final estimate $\Psi_t^n(x)$ and target level $\lambda = 1 - \delta_{\text{mod}}$ because this relationship dominated the observed trends. Re your comments about scalability: the number of observations in this plot is always less-equal to 512 and this number is anti-correlated with $\lambda$. These details aside, we should better communicate that PRB is *much* more expensive to evaluate than the competing methods. We argue this price tag is acceptable since queries are usually far more expensive than the compute cost; however, it is important that we clearly communicate this information.
>
> 1. Feature maps $\boldsymbol{\phi}$ were constructed using a thousand random Fourier features in the manner suggested by [1].
>
> 1. We will clarify that $D$, defined in Assumption A1, denotes the dimensionality of the search space.
>
> 1. We categorized problems as "real" or "synthetic" depending on whether model hyperparameters we fit online to the data. We will improve this terminology in future versions.
>
> #### References
> [1] Sutherland and Sneider, 2015. “On the Error of Random Fourier Features”.

---

> > ### Comment · Reviewer_5ozp · 2024-08-11
> >
> > Thanks for the response and the clarifications. Here are a few follow-up questions, as I'm still confused about some details.
> >
> > > 1. The random variable $Z$ is introduced on line 128 and its definition in the specific context of PRB is given on line (134).
> >
> > Yes, but $Z$'s definition in line 134 is still dependent on some $\mathbf{x} \in \mathcal{X}$, which is not clear. Would the $\mathbf{x}$ at which $Z$ is evaluated correspond to $\mathbf{s}_t$ in Proposition 1?
> >
> > > 3. Runtimes are shown on the right in Figure 3.
> >
> > As far as I understand from reading the caption in Figure 3, those runtimes are calculated from sampling Bernoulli random variables mimicking the $Z$'s which would be calculated for PRB estimates. However, within a BO loop, those $Z$'s would require samples from a GP (followed by multi-start gradient descent) to be evaluated, which apparently is not the case in Figure 3. So, to me, it seems that those runtime estimates are not indicative of the actual runtime of estimating PRB within a BO loop.

---

> > > ### Author Response · Authors · 2024-08-11
> > > **Reply to follow-up questions**
> > >
> > > Thanks again for your input. In future versions, the main text will clearly discussed the following details.
> > >
> > >
> > > **How is $Z$ define?**
> > >
> > > There is one $Z$ for each point at which we evaluated the proposed stopping rule. We evaluated this rule on all previously queried points $\mathbf{x} \in \mathbf{X}_t$ that satisfied a pruning condition
> > > $P(f_t(\mathbf{s}_t)  - f_t(\mathbf{x}) \ge \epsilon)  \le \delta_m$, which checks if $\mathbf{x}$ would satisfy our stopping rule if $\mathbf{s}_t$ was assumed to uniquely maximize $f_t$. If this condition does not hold, then $\mathbf{x}$ cannot satisfy the rule. We also divided $\delta_e^t$ by the number of test points at step $t$ to retain the union bound (line 168). This information is given in Appendix B (as vaguely hinted by line 171), but we should have done a better job here.
> > >
> > >
> > > **Figure 3**
> > >
> > > Only the left subplot uses toy Bernoulli random variables. The middle and right columns show  actual empirical CDFs and runtimes for the PRB rule from our experiments, i.e. $Z$ was sampled by running multi-start gradient ascent on draws of $f_t$. This is why, for example, costs can be seen to increase with dimensionality. Since our initial submission, we have found that these runtimes can be further accelerated by using different concentration inequalities.

---

> > > > ### Comment · Reviewer_5ozp · 2024-08-11
> > > >
> > > > Thanks for the prompt response and for addressing my latest concerns. I'd expect the authors to revise the text considering these and the issues raised by other reviewers. In its current form, there are quite a few (mainly) clarity related issues, but I believe the paper still brings a valuable contribution to the BO community and that its main issues have been addressed in the discussion. Therefore, I'm happy to raise my score.

---

### Author Rebuttal · Authors · 2024-08-06

This post discusses feedback that was common to multiple reviewers. Comments raised by individual reviewers will be addressed in subsequent posts.

As a preliminary remark, we note that reviewers seem to think that our submission makes solid contributions to theory and/or practice. The primary criticism seems to have been that important practical details are either unclear or absent. We are committed to improving our work and hope that you will consider revising your scores if you feel that we have adequately addressed your concerns.


**Presentation**

In writing this paper, we attempted to provide generic explanations for the core components of our algorithm. Based on the feedback we received, it is clear that this ended up being confusing at times and that greater emphasis should have been placed on practical details and how the pieces fit together. We lost the forest for the trees.

We will tighten up the text to ensure that the broader picture stays in focus. Section 3.2 will be reworked to revolve around BO and its notation will be brought in line with the rest of the paper. Practical details will either be clearly presented in the paper or in a suitable appendix. If possible, pseudo-code for the entire BO loop will be added.


**Using gradients to obtain regret indicators**

Reviewers 5ozp and cuEF both pointed out that the text is overly vague about how gradients are used to obtain $\mathbb{1}(r_t(x) \le \epsilon)$. Using function draws gives us access to pathwise derivatives. Throughout, we used multi-start gradient ascent to maximize each path. In detail: we evaluated each path at previously queried locations and (up to) 64 batches of 256 random points, then random gradient ascent starting from the 16 points for each path. If desired, this process can be expedited by using early stopping (line 113).

Per Reviewer 5ozp, the sampled functions will typically be non-convex; so, there is no guarantee that this process succeeds. This is mentioned on line 124, but further discussion should be added (esp. for high-dimensional problems).

---

### Comment · Area_Chair_d3b2 · 2024-08-10
**Discussion period**

Dear reviewers,

Thank you all for your contributions during the review period. Please take a look at the authors' responses and I look forward to a fruitful discussion on this submission in the next few days!

Best regards,
AC for Submission 18743

---

### Decision · Program_Chairs · 2024-09-25

**Decision:**

Accept (poster)

**Comment:**

The paper introduces a novel stopping criterion for Bayesian optimization based on probabilistic regret bounds. The core novelty is a model-based stopping rule that leverages Monte Carlo estimators to decide when to terminate the optimization process, ensuring that the solution meets a predefined regret threshold with high confidence. The theoretical justification is strong, and the empirical results demonstrate the method's potential across a variety of benchmark problems.

Despite the strengths, there are concerns regarding the practical applicability of the method, particularly when modeling assumptions do not hold perfectly. Reviewer cuEF specifically raised concerns about the method's utility in real-world settings, where the alignment between the objective function and the model might not always be guaranteed. These concerns are valid and should be addressed in future revisions to ensure the method's broader applicability.

One point of note is that the idea of using a model to determine when to stop has some precedence in BO literature (e.g., https://arxiv.org/abs/1806.07555), which also deals with using models to estimate stopping conditions while acknowledging model mismatch. This similarity is worth mentioning as it shows that the concept of model-based stopping has been explored in different contexts.

During the discussion phase, reviewers acknowledged the technical contribution of the paper, though some practical concerns remain. Given these factors, the paper is leaning towards acceptance, provided there is room for further refinement, particularly in clarifying its practical significance and addressing concerns around real-world deployment.